# Oversampling to Repair Bias and Imbalance Simultaneously

**Martin Hirzel**[1]  **Parikshit Ram**[1]

[1]IBM Research, USA

**Abstract**  Both group bias and class imbalance occur when instances with certain characteristics are under-represented in the data. Group bias causes estimators to be unfair and class imbalance causes estimators to be inaccurate. Oversampling ought to address both kinds of under-representation. Unfortunately, it is hard to pick a level of oversampling that yields the best fairness and accuracy for a given estimator. This paper introduces Orbis, an oversampling algorithm that can be precisely tuned for both fairness and accuracy. Orbis is a pre-estimator bias mitigator that modifies the data used to train downstream estimators. This paper demonstrates how to use automated machine learning to tune Orbis along with the choice of estimator that follows it and empirically compares various approaches for blending multiple metrics into a single optimizer objective. Overall, this paper introduces a new bias mitigator along with a methodology for training and tuning it.

## 1 Introduction

Machine learning often suffers from the twin problems of group bias and class imbalance. In a classification setting, *class imbalance* occurs when the number of instances with one class label is smaller than with another class label. Class imbalance has long been recognized as a problem, because many models perform poorly for minority classes, and in many applications, the cost of misprediction is unequal across classes [8, 15]. One definition for *group bias* is that instances in one group experience a smaller ratio of favorable outcomes than another group [12]. Here, a *group* comprises all instances for which a protected attribute such as race or gender has a certain value, or a protected attribute such as age falls on one side of a certain threshold. And an *outcome* is the prediction target of the instance, in this paper, a class label. Group bias is increasingly recognized as a problem because it can cause ethical, legal, reputational, and financial harm. Often, a group experiencing bias is also a minority group, i.e., it is under-represented in the training data.

Both problems, group bias and class imbalance, involve subsets of instances being under-represented. Having fewer samples makes it harder for models to generalize. Sub-dividing data by intersecting groups and classes further exacerbates this limited-data problem. Fortunately, there are algorithms for mitigating group bias (e.g. reject option classification [19]) and class imbalance (e.g. Smote [8]). However, this paper shows that mitigating either goal separately can harm the other goal; for example, when Smote reduces the class imbalance of a dataset, that can exacerbate its group bias. Furthermore, it is not clear how much to mitigate imbalance or bias in the data to achieve the desired effect in estimators trained from that data. We refer to the amount of data mitigation for imbalance or bias as the *repair level*. Since the effect of repair levels on estimators is unpredictable, we argue they should be tuned automatically, as hyperparameters.

This paper introduces the Orbis algorithm, which stands for Oversampling to Repair Bias and Imbalance Simultaneously. Orbis extends Smote [8] to repair for both objectives such that the repair level for each can be precisely controlled via two hyperparameters. In experiments across 12 datasets, Orbis performs well compared to 5 other imbalance mitigators and 9 other bias mitigators from prior work. Since Orbis is designed with automated machine learning (AutoML) in mind, this paper also elaborates on an estimator evaluation workflow for that setting. We define 5 approaches for blending metrics for accuracy and fairness into a single objective and empirically

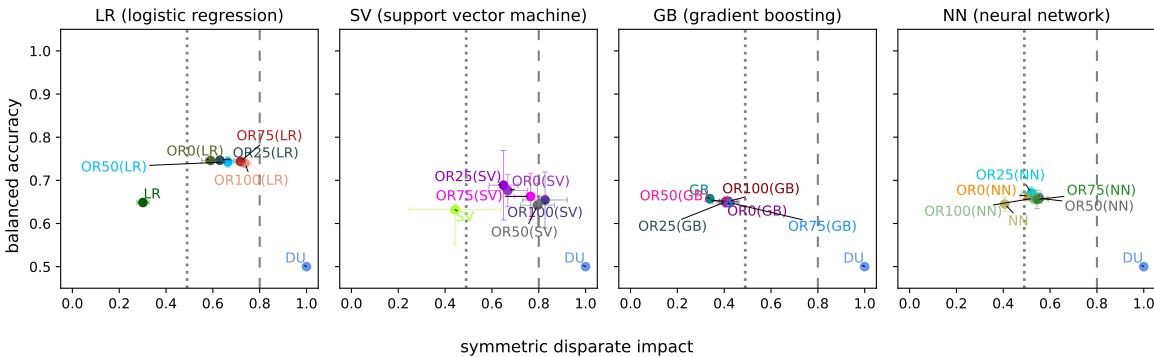

Figure 1: The effect of repair levels and estimators. For both balanced accuracy and symmetric disparate impact (a fairness metric), higher is better and the ideal value is 1. OR$\delta_{\text{bias}}$ denotes ORBIS with imbalance repair level 80% and bias repair level $\delta_{\text{bias}}$. DU is the dummy classifier.

compare what effect each approach has on AutoML performance. The code for ORBIS (along with dataset fetchers, wrappers for other mitigators, etc.) is open-source (https://github.com/ibm/lale commit e0b4f44 and https://test.pypi.org/project/lale/0.7.8.post2306082350/).

## 2 Motivation and Problem Statement

To motivate the need for tunable repair level hyperparameters, this section starts with an example demonstrating that the effect of imbalance and bias repair on estimators can be unpredictable.

Figure 1 shows results for ORBIS with different repair levels and estimators on the meps20 dataset [2]. The y-axis shows balanced accuracy, i.e., the average per-class recall, where higher values are more accurate. The x-axis shows symmetric disparate impact, where higher values are more fair. Disparate impact is the ratio of the favorable rate of the unprivileged group to the favorable rate of the privileged group [12]. It can be computed either using labels predicted by an estimator trained on the data as done for Figure 1 or using ground-truth labels. Symmetric disparate impact is the same as disparate impact for values below one and its reciprocal otherwise. Each point is an average of six runs (two repeats of 3-fold cross validation) and the error bars show one standard deviation. The dashed lines at symmetric disparate impact 0.8 indicate the 80% rule [12] and the dotted lines show the disparate impact computed using ground-truth labels of the dataset. The dummy classifier, which always predicts the majority class, is always at the bottom right, with the best symmetric disparate impact of 1 and the worst balanced accuracy of 0.5.

The leftmost plot in Figure 1 shows that for logistic regression, the highest repair level (OR100) yields both the best accuracy and best fairness. Moving to the next plot, for the support vector machine, repair levels up to 50% improve both metrics, but above that, higher bias repair causes better fairness at the expense of worse accuracy. For gradient boosting, repair hardly affects either metric. Finally, for the neural network, repair has a slightly larger effect than for gradient boosting, but the effect is still too small to draw conclusions. Overall, these results show that repair levels can make a big difference and their effect is hard to predict a-priori.

Next, we will look at an example that motivates the need to repair imbalance and bias simultaneously, because repairing either separately can make the other worse. Consider a binary protected attribute whose value can be either unprivileged (0) or privileged (1) and a binary target label whose value can be either unfavorable (0) or favorable (1). This divides a dataset into four intersections of sizes $o_{00}$ (unprivileged unfavorable), $o_{01}$ (unprivileged favorable), $o_{10}$ (privileged unfavorable), and $o_{11}$ (privileged favorable). Define the original (before repair) class imbalance $o_{\text{ci}}$ and group bias $o_{\text{di}}$ (measured by disparate impact) as

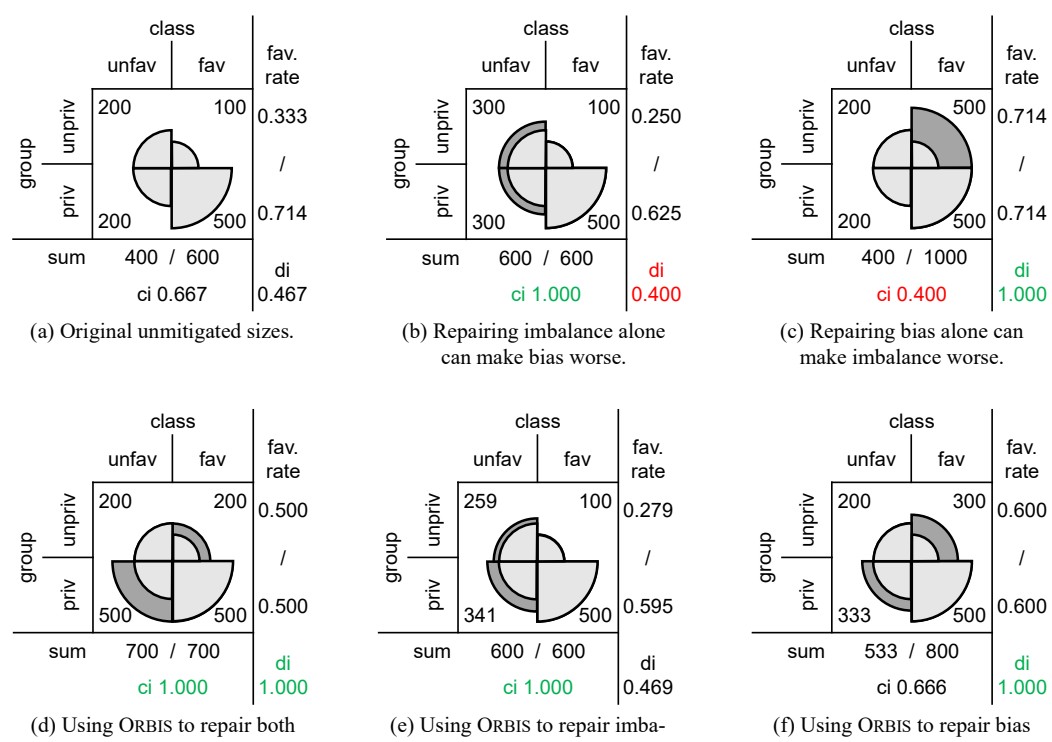

Figure 2: Different oversampling sizes for repairing imbalance and/or bias. Light gray represents original data and dark gray data added by oversampling. The numbers next to "sum" show total sizes of both classes; dividing them yields class imbalance ci. The numbers below "fav. rate" show favorable rates of both groups; dividing them yields disparate impact di.

$$o_{ci} = \frac{o_{00} + o_{10}}{o_{01} + o_{11}} \qquad \bigwedge \qquad o_{di} = \frac{o_{01}/(o_{01} + o_{00})}{o_{11}/(o_{11} + o_{10})}, \qquad (1)$$

where the numerator of $o_{di}$ is the favorable rate for the unprivileged group and the denominator is the favorable rate for the privileged group [12].

Figure 2 illustrates different choices for oversampling the intersections of the data to new sizes $n_{00}$, $n_{01}$, $n_{10}$, and $n_{11}$. The starting point in Figure 2(a) is a dataset with $o_{00} = 200$, $o_{01} = 100$, $o_{10} = 200$, and $o_{11} = 500$. This dataset has a class imbalance of $o_{ci} = 0.667$ and a group bias of $o_{di} = 0.467$. The ideal values for both class imbalance and group bias is 1. Figure 2(b) shows the effect of oversampling to repair class imbalance while being oblivious to group bias, i.e., the effect of using an algorithm such as Smote [8] out of the box. Unfortunately, this makes bias worse, reducing di from 0.467 to 0.400. Similarly, Figure 2(c) shows that a naive bias repair algorithm that only oversamples the unprivileged favorable intersection would make imbalance worse. In contrast, Figure 2(d) shows how Orbis can repair both imbalance and bias simultaneously. While this is useful, Figure 1 demonstrated that the highest repair level for the dataset does not always yield the best metrics for an estimator trained on that data. Therefore, Orbis lets users tune imbalance and bias separately. Figure 2(e) shows a solution that repairs class imbalance while carefully controlling the new group bias to be no different from the original. Similarly, Figure 2(f) shows a solution that repairs group bias while keeping class imbalance unchanged.

We can control imbalance and bias in a more fine-grained manner than the examples in Figure 2. Let $\delta_{imbalance} \in [0, 1]$ and $\delta_{bias} \in [0, 1]$ be continuous hyperparameters controlling the repair level for class imbalance and group bias, respectively. Further, denote by $n_{ci}$ the new class imbalance and

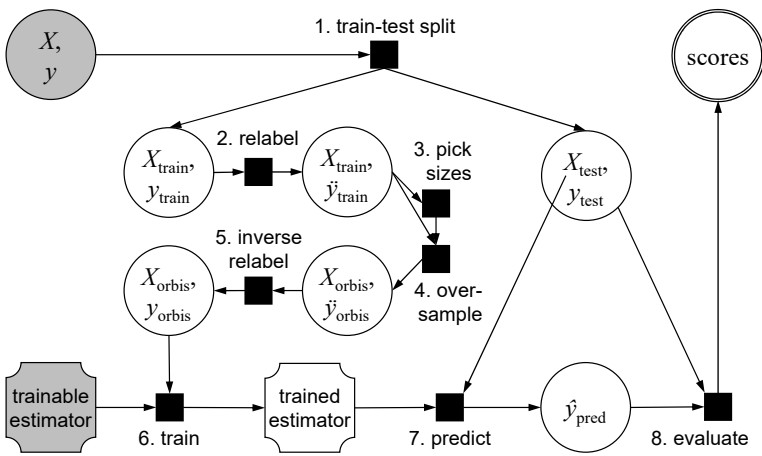

Figure 3: Overview of the Orbis algorithm in the context of an estimator evaluation workflow.

by $n_{di}$ the new disparate impact after oversampling, as computed from the new sizes $n_{00}$, $n_{01}$, $n_{10}$, and $n_{11}$. The problem statement is to pick these new sizes and oversample to satisfy the following two constraints:

$$n_{ci} = o_{ci} + \delta_{imbalance}(1 - o_{ci}) \qquad \bigwedge \qquad n_{di} = o_{di} + \delta_{bias}(1 - o_{di}) \qquad (2)$$

## 3 Orbis Algorithm

The core of Orbis consists of computing the desired intersection sizes and then oversampling the data accordingly. That said, there are also hard-learned lessons for the workflow around this core that may trip up the unwary [28]. Therefore, Figure 3 shows Orbis along with a recommended workflow for evaluating estimators, suitable for AutoML. The rest of this section explains the steps from Figure 3 given a matrix $X$ of features and a vector $y$ of binary class labels for each instance.

**Step 1: Train-test split**. This step partitions $X$, $y$ into $X_{train}$, $y_{train}$ and $X_{test}$, $y_{test}$. One potential pitfall with oversampling is that the test data may contain a synthetic instance obtained by oversampling a real instance or vice versa, causing over-fitting [28]. This must be prevented by only applying oversampling to the training data, i.e., experiments must first split the data and only then perform oversampling. To ensure this by construction, we implemented Orbis as a meta-estimator [4] that takes the downstream estimator as an argument, oversamples $X_{train}$, $y_{train}$ during fitting, and passes an unmodified $X_{test}$ through to the downstream estimator when predicting. The meta-estimator itself can serve as an argument to an automated hyperparameter tuning tool that uses a cross-validation split. We recommend stratifying splits by both groups and classes [17], because in highly imbalanced data, a non-stratified split risks some intersections of groups and classes being tiny or even empty.

**Step 2: Relabel**. To oversample specific intersections of groups and classes, it is useful to have explicit labels for these intersections. Therfore, this step changes $X_{train}$, $y_{train}$ into $X_{train}$, $\ddot{y}_{train}$, where $\ddot{y}_{train}$ contains *diaeresis labels*. (The word diaeresis refers to the two dots above the $y$; it comes from the Greek word for separation, since these labels induce a separation of the data.) Let *get_group* : $x_i \rightarrow \{0, 1\}$ be a function that indicates, for a given row $x_i$ representing one instance, whether that instance belongs to the unprivileged (0) or privileged (1) group. For example, *get_group* might retrieve a numeric *age* attribute and apply a threshold to group instances into young or old. Similarly, let *get_class* : $y_i \rightarrow \{0, 1\}$ be a function that maps labels to unfavorable (0) or favorable (1) classes. The diaeresis labels are simply pairs $\ddot{y}_i = \langle get\_group(x_i), get\_class(y_i) \rangle \in \{0, 1\}^2$. The

*get_class* function must be invertible whereas *get_group* does not need to be invertible; in fact, the input to *get_group* may comprise multiple, non-binary, or even continuous protected attributes.

**Step 3: Pick sizes.** Given $X_{\text{train}}, \ddot{y}_{\text{train}}$, this step computes the desired new sizes $n_{00}, n_{01}, n_{10}$, and $n_{11}$. The diaeresis labels $\ddot{y}_{\text{train}}$ induce a partition on the instances into subsets that share the same label. ORBIS computes the original sizes $o_{00}, o_{01}, o_{10}$, and $o_{11}$ of these intersections, and from these, computes the original class imbalance $o_{\text{ci}}$ and disparate impact $o_{\text{di}}$ (Equation 1). Next, it uses Equation 2 to compute the desired new class imbalance $n_{\text{ci}}$ and disparate impact $n_{\text{di}}$ based on hyperparameters $\delta_{\text{imbalance}}$ and $\delta_{\text{bias}}$. Without loss of generality, assume $n_{\text{ci}} \leq 1$ (if not, swap classes) and $n_{\text{di}} \leq 1$ (if not, swap groups). The desired solution needs to satisfy two equations:

$$\frac{n_{00} + n_{10}}{n_{01} + n_{11}} = n_{\text{ci}} \qquad \bigwedge \qquad \frac{n_{01}/(n_{01} + n_{00})}{n_{11}/(n_{11} + n_{10})} = n_{\text{di}} \qquad (3)$$

Given four unknowns $(n_{00}, n_{01}, n_{10}, n_{11})$, these equations permit many possible solutions. Since $n_{\text{ci}} \leq 1 \wedge n_{\text{di}} \leq 1$, we can eliminate one unknown by simply setting $n_{11} = o_{11}$. Next, we will strive to minimize oversampling the intersection of the unprivileged group with members receiving unfavorable class labels, because it is most likely to exemplify the kind of bias the algorithm is intended to mitigate. To do this, we will find the minimum $n_{00}$ for which solving the above equation satisfies $n_{00} \geq o_{00} \wedge n_{01} \geq o_{01} \wedge n_{10} \geq o_{10} \wedge n_{11} \geq o_{11}$. Having eliminated two unknowns, $n_{11}$ and $n_{00}$, all that remains is to solve for the remaining two unknowns, $n_{01}$ and $n_{10}$. It can be shown that the equations above imply $n_{10} = \frac{1}{2}(\sqrt{b^2 - 4c} - b)$, where $b = n_{00} + n_{11} - n_{11}n_{\text{ci}} - n_{11}n_{\text{di}}$ and $c = n_{00}n_{11} + n_{11}n_{11}n_{\text{ci}}n_{\text{di}} - n_{00}n_{11}n_{\text{ci}}n_{\text{di}} - n_{11}n_{11}n_{\text{ci}} - n_{00}n_{11}n_{\text{di}}$. And finally, $n_{01} = \frac{n_{00}}{n_{\text{ci}}} + \frac{n_{10}}{n_{\text{ci}}} - n_{11}$. See Appendix D for the detailed size selection scheme.

After picking the sizes, ORBIS obtains the final numbers by reversing the swap of classes and groups, if any, that was needed to ensure $n_{\text{ci}}$ and $n_{\text{di}}$ are at most one. This has the effect that ORBIS repairs imbalance or bias symmetrically for whichever class or group exhibits it in the data.

**Step 4: Oversample.** Given $X_{\text{train}}, \ddot{y}_{\text{train}}$ and the desired new sizes $n_{00}, n_{01}, n_{10}$, and $n_{11}$, this step creates more balanced training data $X_{\text{orbis}}, \ddot{y}_{\text{orbis}}$. This step applies SMOTE [8], which stands for Synthetic Minority Oversampling Technique, to each of the four intersections separately. While the intersection has not yet reached its new desired size, repeat the following:

(i) Randomly choose a non-synthetic instance $r$ from the given intersection to oversample.
(ii) Find the $k$ non-synthetic instances that are nearest neighbors of $r$, and randomly choose an instance $v$ among them that is in the same intersection as $r$.
(iii) Randomly choose a number $\varphi$ between 0 and 1.
(iv) Create a new synthetic instance $s = r + \varphi(v - r)$.

One potential problem is that the group of a synthetic instance $s$ might differ from that of the real instance $r$ it was derived from. This can be avoided by ensuring that the function *get_group* satisfies $get\_group(r) = get\_group(r + \varphi(v - r))$ for any two instances $r$ and $v$ with $get\_group(r) = get\_group(v)$ and $0 \leq \varphi \leq 1$. Another technical issue is that ORBIS should handle categorical features; for instance, protected attributes are often categorical. We handle this with the SMOTE-NC and SMOTE-N algorithms [8] implemented in the imbalanced-learn library [23].

**Step 5: Inverse relabel.** This step changes $X_{\text{orbis}}, \ddot{y}_{\text{orbis}}$ into $X_{\text{orbis}}, y_{\text{orbis}}$. It simply retrieves the class component of the diaeresis label $\ddot{y}_{\text{orbis}}$ and applies the inverse of the *get_class* function.

**Step 6: Train.** Given the oversampled training data $X_{\text{orbis}}, \ddot{y}_{\text{orbis}}$ and the trainable downstream estimator, this step creates the trained estimator. Recall that the trainable downstream estimator is itself an argument to the meta-estimator. In fact, an AutoML tool can even treat it as a hyperparameter, to be tuned for automated algorithm selection. In our experiments, the downstream estimator

is actually a pipeline comprising first an ordinal encoder for categorical features (forwarding continuous features as-is), followed by one of four scikit-learn [4] operators: logistic regression, support vector machine, gradient boosting, or a multi-layer perceptron neural network classifier.

**Step 7: Predict.** This step applies the trained estimator on $X_{\text{test}}$ to obtain predictions $\hat{y}_{\text{pred}}$.

**Step 8: Evaluate.** The last and final step of the workflow from Figure 3 computes scores. Unlike accuracy metrics that only require ground-truth labels $y_{\text{test}}$ and predicted labels $\hat{y}_{\text{pred}}$, fairness metrics typically also require $X_{\text{test}}$ to inspect protected attributes. This step can compute multiple metrics separately, such as symmetric disparate impact (DI) and balanced accuracy (BA) serving as the x-axis and y-axis in Figure 1. In addition, for use with a single-objective optimizer, this step can also compute blended metrics. Since there is no consensus on the best approach for blending metrics, this paper considers a variety of approaches:

- Arithmetic mean, $\text{AM} = \frac{\text{BA+DI}}{2}$, is the most familiar and straight-forward to explain.
- Geometric mean, $\text{GM} = \sqrt{\text{BA} \cdot \text{DI}}$, quantifies the area of Pareto dominance in the scatter plot.
- Harmonic mean, $\text{HM} = \frac{2 \cdot \text{BA} \cdot \text{DI}}{\text{BA+DI}}$, also encourages a larger area of Pareto dominance while tolerating differences in scale between the component metrics better than geometric mean does.
- Hard threshold, $\text{HT} = \begin{cases} \frac{\text{DI}}{2 \cdot \tau} & \text{if DI} < \tau \\ \text{BA} & \text{otherwise} \end{cases}$, focuses exclusively on DI when DI is below a fairness threshold $\tau$, and on BA above.
- Soft threshold, $\text{ST} = \begin{cases} \text{BA} \cdot (\frac{\text{DI}}{\tau})^4 & \text{if DI} < \tau \\ \text{BA} & \text{otherwise} \end{cases}$, focuses mostly on DI when DI is below a fairness threshold $\tau$, but also rewards improvements to BA in that regime a little, and focuses on BA when DI is above the threshold $\tau$.

We chose to formulate this as a single-objective hyperparameter optimization or HPO problem by considering different strategies of combining the predictive and fairness performance. This could also have been posed as multi-objective HPO. But our HPO problem is more constrained since based on the 80% rule of the US Equal Employment Opportunity Commission, we really desire disparate impact to be above 80% [12]. That does not directly fit into usual multi-objective HPO solvers, while the combined objectives HT and ST support it directly by setting $\tau = 0.8$. Furthermore, we often need to find a single solution (which our scheme produces) instead of requiring the user to select from a (potentially large) set of Pareto-optimal solutions.

## 4 Empirical Study

This section empirically studies three research questions:
**RQ1.** How do different imbalance mitigators affect fairness and predictive performance?
**RQ2.** How do different bias mitigators affect fairness and predictive performance?
**RQ3.** How do different approaches for blending metrics affect single-objective AutoML?
See the supplemental material for more details on our study.

**Datasets.** We consider 12 binary classification datasets (4 from AIF360 [2] and 8 from OpenML [33]) shown in Table 1. In only 3 of the 12 datasets, the disparate impact would be considered as fair (with $o_{\text{di}}$ above 0.8), and only 2 datasets are relatively balanced (with $o_{\text{ci}}$ around 0.9), highlighting the need to study bias in conjunction with class imbalance. While MEPS 19 and MEPS 20 are different datasets with no overlap, their imbalance and fairness characteristics are similar. On the other hand, even though COMPAS Violent is a subset of COMPAS, their characteristics are quite different: COMPAS Violent is significantly less balanced but has better base disparate impact.

For each dataset and each configuration, we perform a total of six runs, comprising two repeats of 3-fold cross validation. Figures 4, 5, and 6 show the results. Each figure has 12 subfigures, one

Table 1: Datasets in ascending order of #ROWS. Columns $o_{ci}$ and $o_{di}$ show original class imbalance and disparate impact. Datasets marked with † are from AIF360, the remainder are from OpenML.

| DATASET | DESCRIPTION | PROTECTED ATTRIBUTE | #ROWS | $o_{ci}$ | $o_{di}$ |
|---|---|---|---|---|---|
| Ricci | Fire department promotion exam results | race | 118 | 0.90 | 0.50 |
| TAE | University teaching assistant evaluation | TA-native-speaker | 151 | 0.50 | 1.74 |
| Credit-g | German bank data quantifying credit risk | sex,age | 1,000 | 0.43 | 0.75 |
| Titanic | Survivorship of Titanic passengers | sex | 1,309 | 0.62 | 0.26 |
| COMPAS Violent† | Correctional offender violent recidivism | sex,race | 3,377 | 0.21 | 0.82 |
| COMPAS† | Correctional offender recidivism | sex,race | 5,278 | 0.89 | 0.69 |
| SpeedDating | Speed dating experiment at business school | samerace,imp-samerace | 8,378 | 0.20 | 0.85 |
| Nursery | Slovenian nursery school application results | parents | 12,960 | 0.45 | 0.46 |
| MEPS 19† | Utilization results from Panel 19 of MEPS | RACE | 15,830 | 0.21 | 0.49 |
| MEPS 20† | Same as MEPS 19 except for Panel 20 | RACE | 17,570 | 0.21 | 0.49 |
| Bank | Portuguese bank subscription predictions | age | 45,211 | 0.13 | 0.84 |
| Adult | 1994 US Census salary data | sex,race | 48,842 | 0.31 | 0.23 |

for each dataset, sorted by size. The axes, metrics, error bars, and dotted and dashed vertical lines have the same meaning as in Figure 1. Symmetric disparate impact is consistently more noisy (horizontal error bars) than balanced accuracy (vertical error bars), an effect that would be even more visible if both axes used the same scale. Larger datasets tend to have smaller error bars.

**RQ1: How do different imbalance mitigators affect fairness and predictive performance?** Figure 4 shows results for several mitigators that either repair only class imbalance or use rebalancing to repair group bias. As expected, SMOTE [8], and its variants SMOTEN/SMOTENC depending on the data, improve balanced accuracy over unmitigated LR significantly in 4 datasets, while never being significantly worse. ORBIS (with $\delta_{imbalance} = 0.8$ and $\delta_{bias} = 1$) improves disparate impact over SMOTE for most datasets while maintaining the same level of balanced accuracy. FOS [10] and Fair-SMOTE [6] usually perform similarly to ORBIS, but ORBIS has the additional advantage of being tunable, as shown in Figure 1. Reweighing [18] usually does worse than the oversampling based approaches, but excels for creditg and nursery. Undersampling-multivariate [32] generally sacrifices more accuracy than oversampling based approaches but excels at compas. Overall, Figure 4 shows that even without hyperparameter tuning, ORBIS is very competitive.

**RQ2: How do different bias mitigators affect fairness and predictive performance?** Figure 5 compares ORBIS (using $\delta_{imbalance} = 0.8$ and $\delta_{bias} = 1$) against nine other bias mitigators from AIF360 [2] (using their default hyperparameters). In general, different mitigators trade-off predictive performance and bias to different degrees, sometimes tracing out a Pareto frontier. Even without hyperparameter tuning, ORBIS is Pareto-optimal for most of the datasets, more often than any other mitigator, since it tends to yield high balanced accuracy while also improving fairness. RO (reject-option classification [19]) is also often a front-runner. However, RO sometimes degenerates to perform like a dummy classifier, with optimal fairness but low accuracy. There is no "one size fits all" for bias mitigators, and it is important to try available options rigorously.

**RQ3: How do different approaches for blending metrics affect single-objective AutoML?** Figure 6 shows results from using Hyperopt [3] in Lale [1] to jointly tune the hyperparameters and select the downstream estimator passed to ORBIS. We let Hyperopt tune $\delta_{imbalance}$ and $\delta_{bias}$, both in the range from 0 to 1, while selecting among a choice between scikit-learn's [4] logistic regression, support vector machine, gradient boosting, or a multi-layer perceptron neural network. For each blending approach from Step 8 of Section 3, we launch 3 Hyperopt runs with randomly shuffled data, where each run has 20 trials, and each trial performs 3-fold cross validation. The scatter plot shows the average result of the best configuration found, with error bars for standard deviation across the 3 runs. Overall, geometric mean is usually effective at finding an ORBIS configuration that

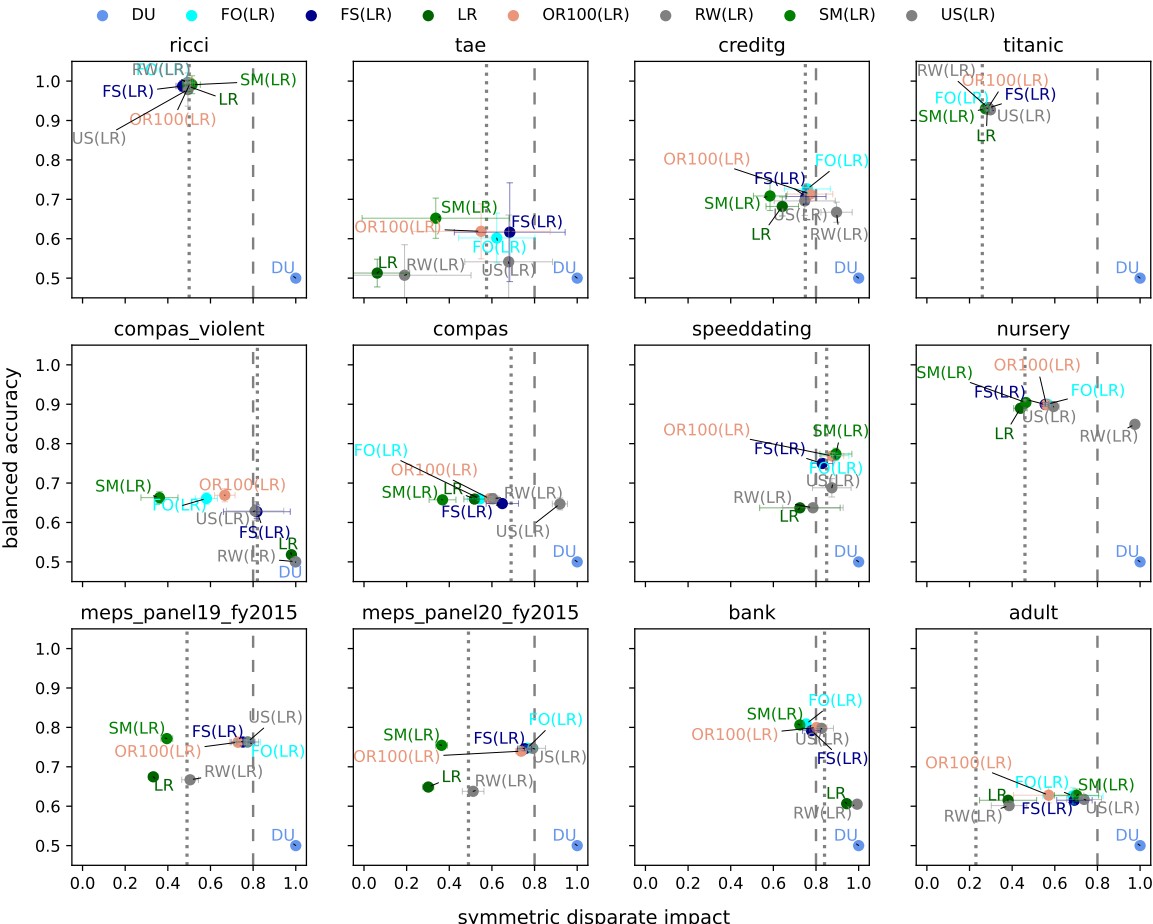

Figure 4: Comparing class imbalance mitigators. DU is the dummy estimator, FO is FOS [10], FS is Fair-Smote [6], LR is unmitigated logistic regression, OR100 is Orbis with $\delta_{\mathrm{imbalance}} = 0.8$ and $\delta_{\mathrm{bias}} = 1$, RW is reweighing [18], SM is Smote [8], and US is undersampling-multivariate [32].

does well on both axes. The threshold approaches sometimes do well at approaching or surpassing a disparate impact of 0.8, but struggle from noise with some datasets and degenerate to perform like dummy for some others. Innovation in reining in noise in the metrics across folds could make AutoML more effective at mitigating bias. In the meantime, we recommend the geometric mean.

## 5 Related Work

The literature on class imbalance mitigators is extensive. Interested readers can find an excellent survey in He and Garcia, who discuss oversampling, undersampling, cost-sensitive learning, etc. [15]. Smote [8] is one of the most popular class imbalance mitigators. In an empirical study of oversamplers by Santos et al. [28], Smote consistently performs among the top of 12 class imbalance mitigators, and in fact, 10 of the other mitigators they studied extend Smote. Imbalanced-learn is an open-source library of imbalance mitigators [23]. Unlike our paper, none of the above works address group bias or AutoML. AutoBalance combines 12 class imbalance mitigators with AutoML, but does not discuss group bias [29]. BalaGen explores class imbalance correction with both oversampling and undersampling hyperparameters for text data, but does not discuss group bias [31].

A few works adapt class imbalance mitigators for mitigating group bias. Fair-Smote [6] applies Smote to oversample all non-majority intersections of groups and classes to the size of the majority, thereby allowing only the highest level of repair for imbalance and bias. As demonstrated in

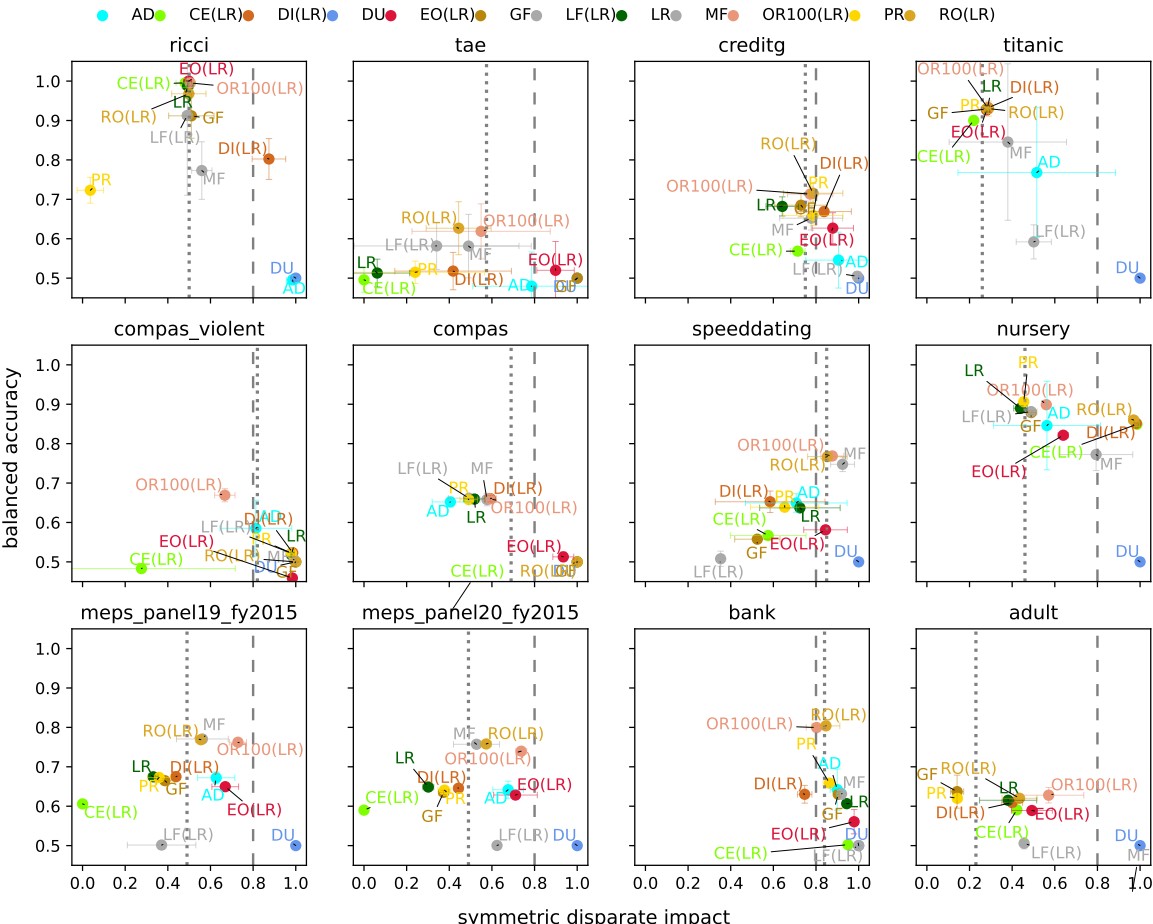

Figure 5: Comparing group bias mitigators. AD is adversarial debiasing [37], CE is calibrated equalized-odds post-processing [27], DI is disparate impact remover [12], DU is the dummy estimator, EO is equalized-odds post-processing [14], GF is gerry-fair classifier [22], LF is learning fair representations [36], LR is unmitigated logistic regression, MF is meta-fair classifier [5], OR100 is ORBIS with $\delta_{\text{imbalance}} = 0.8$ and $\delta_{\text{bias}} = 1$, PR is prejudice remover [20], and RO is reject-option classification [19].

Section 2, that is not always the best option. Furthermore, it can lead to unnecessarily high amounts of synthetic data. FOS [10] also extends SMOTE for bias mitigation. It takes a slightly different approach, internally class-balancing each group. ORBIS reduces to FOS when the repair level is set to the highest value for both imbalance and bias, but FOS does not consider intermediate levels of repair. Reweighing [18] is a group bias mitigator that, like ORBIS, effectively changes the total "size" of certain data subsets. It does not address class imbalance. Undersampling-multivariate can mitigate group bias, class imbalance, or both together, but does not explore repair level hyperparameters [32]. Cost-sensitive learning can repair class imbalance via a loss function. The FBI-loss repairs either class imbalance or group bias, depending on how it is instantiated [13]. LDAM$_{\text{reg}}$ adds a loss function for repairing class imbalance to a regularization term for group bias [30]. The FBI-loss and LDAM$_{\text{reg}}$ have been demonstrated only with neural networks; in contrast, this paper demonstrates ORBIS with neural networks as well as other base estimators.

Fairness-aware AutoML [34] incorporates fairness either as (i) an objective alongside the predictive performance with multi-objective hyperparameter optimization [21, 25], or (ii) as a constraint for a given threshold [24, 26]. However, neither studies the class imbalance and bias mitigation

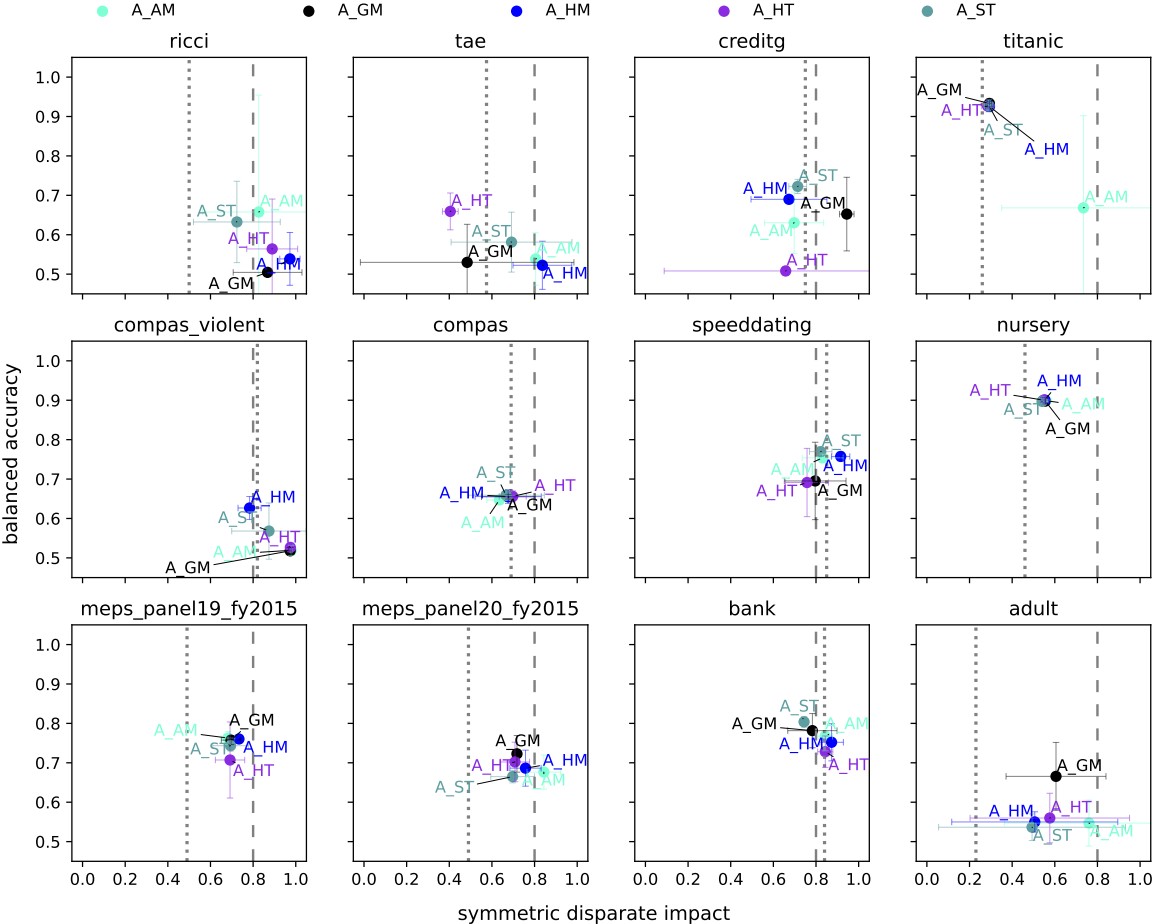

Figure 6: Comparing different approaches for blending metrics into an AutoML objective. AM, GM, and HM are arithmetic, geometric, and harmonic mean. HT and ST are hard and soft threshold with $\tau = 0.8$.

simultaneously as we do. Feffer et al. [11] explore AutoML with bias mitigation and ensembles, but do not consider imbalance mitigation. Some fairness-aware AutoML results [7, 9] seem to indicate that one can achieve a better accuracy-bias tradeoff by tuning model hyperparameters than by using bias mitigators, but Wu and Wang [35] provide counter-examples for that. None of them try to optimize over the hyperparameters of the (bias as well as imbalance) mitigators as we do. Our current evaluation considers a blended metric for hyperparameter optimization, studying the effect of combining accuracy and bias in different ways. Given the search space definition we propose, we can consider a constrained multi-objective hyperparameter optimizer.

This paper uses 12 datasets for evaluation. After writing this paper, we added more datasets to create an open-source suite of 20 functions for fetching dataset and adding fairness metadata [16].

## 6 Conclusion

This paper demonstrates that repairing bias or imbalance separately can harm imbalance or bias, respectively. Furthermore, full mitigation is not always best, and indeed, it is difficult to decide ahead of time which repair level to apply. Next, this paper introduces ORBIS, an algorithm that mitigates imbalance and bias simultaneously, and that allows the user to choose the exact repair levels for both. This paper discusses how to use ORBIS in an AutoML context, and includes an extensive experimental evaluation.

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

## A  Broader Impact Statement

After careful reflection, the authors have determined that this work presents no notable negative impacts to society or the environment.


## C  Limitations

As described in this paper, ORBIS requires binary class labels. ORBIS supports multiple non-binary protected attributes, but it turns them into a single binary protected attribute and performs bias repair only for the resulting binary groups (Step 2 of the algorithm in Section 3). Our empirical evaluation uses only binary classification datasets, some of which have multiple protected attributes, and some have non-binary or even continuous protected attributes. Extending ORBIS to cases with more than two classes or groups would lead to more unknowns in the equations for picking sizes (Step 3 of the algorithm in Section 3). We believe this is solvable but leave it to future work.

ORBIS is guaranteed to always yield the requested levels of imbalance and bias in the training data, modulo rounding effects from non-integer numbers of samples. However, it cannot guarantee the desired accuracy and fairness of the trained estimator. For instance, in Figure 2, ORBIS works less well for gradient boosting for the meps20 dataset. We conjecture that this may be because later boosting rounds sub-samples data, reducing effects of pre-estimator imbalance correction. This motivates using ORBIS together with automated estimator selection, which this paper demonstrates.

## D Picking sizes for repair

With $n_{11} = o_{11}$, for a size $n$, let us define the following terms:

$$b(n) \triangleq n + n_{11} - n_{11}n_{ci} - n_{11}n_{di}, \quad c(n) \triangleq nn_{11} + n_{11}^2 n_{ci}n_{di} - nn_{11}n_{ci}n_{di} - n_{11}^2 n_{ci} - nn_{11}n_{di}. \quad (4)$$

This is exactly the definition of $b$ and $c$ in Section 3. Then, we search for the size $n_{00}$, and consequently $n_{10}, n_{01}$ as follows, given $n_{11}, n_{ci}, n_{di}$ and the old sizes $o_{00}, o_{01}, o_{10}, o_{11}$:

- For $n \in [o_{00}, (o_{00} + o_{01} + o_{10} + o_{11})]$:

  - Compute $b(n)$ and $c(n)$ as in equation 4
  - If $b^2(n) < 4c(n)$ continue with next iteration
  - Set $n_{10} \leftarrow \frac{1}{2}(\sqrt{b^2(n)) - 4c(n)} - b(n))$
  - Set $n_{01} \leftarrow \frac{n_{00}}{n_{ci}} + \frac{n_{10}}{n_{ci}} - n_{11}$
  - If $n_{10} < o_{10}$ or $n_{01} < o_{01}$ continue with next iteration
  - $n_{00} \leftarrow n$ and break from the loop

With this procedure, we have selected the smallest $n_{00}$ that satisfies the desired repair level constraints (equation 3). We try to select the smallest $n_{00}$ to ensure that the least amount of data is synthetically generated since that can lead to statistical and computational issues.

We arrived at the formulas for $b$ and $c$ as follows. First, we solved the $n_{ci}$ equation for $n_{01}$. Next, we substituted that formula for $n_{01}$ into the $n_{di}$ equation. Then, we rewrote the resulting formula into the standard form of a quadratic equation for $n_{10}$. Then we substituted $b$ and $c$ for the appropriate terms in the quadratic equation. After computing $b$ and $c$, we used those to compute $n_{10}$. Finally, we substituted that solution for $n_{10}$ into the rewritten $n_{ci}$ equation to obtain $n_{01}$.

## E Different fairness metrics

All experimental results in the main paper use symmetric disparate impact, which is the same metric that ORBIS uses internally for picking subset sizes. All of our experimental runs also record 3 other fairness metrics, and Figures 7–9 report the results for comparison to Figure 4. Let $g$ refer to the binary group. The four fairness metrics are:

- Symmetric disparate impact, $\text{SDI} = \begin{cases} \text{DI} & \text{if DI} \leq 1 \\ \frac{1}{\text{DI}} & \text{otherwise} \end{cases}$,
  based on the disparate impact $\text{DI} = \Pr(\hat{y} = 1 \mid g = 0)/\Pr(\hat{y} = 1 \mid g = 1)$.
  Disparate impact is the rate of positive outcomes for the unprivileged group divided by the rate of positive outcomes for the privileged group. It is non-negative and its ideal value is 1. Figure 4 in the main paper shows results for symmetric disparate impact.

- Statistical parity difference, $\text{SPD} = \Pr(\hat{y} = 1 \mid g = 0) - \Pr(\hat{y} = 1 \mid g = 1)$.
  Statistical parity difference is similar to disparate impact, using subtraction instead of division of the rates of positive outcomes. It is in $[-1, 1]$ and its ideal value is 0. Figure 7 shows the results. Overall, the qualitative conclusions for statistical parity difference are similar to those for symmetric disparate impact. ORBIS is effective at repairing statistical parity difference.

- Equal opportunity difference, $\text{EOD} = \Pr(\hat{y} = 1 \wedge y = 1 \mid g = 0) - \Pr(\hat{y} = 1 \wedge y = 1 \mid g = 1)$.
  Equal opportunity difference is the difference of the true positive rate for the unprivileged and privileged group. It is in $[-1, 1]$ and its ideal value is 0. Figure 8 shows the results. Since ORBIS, Fair-SMOTE, FOS, and undersampling-multivariate all optimize for disparate impact, there are some datasets where none of them have the desired effect on equal opportunity difference.

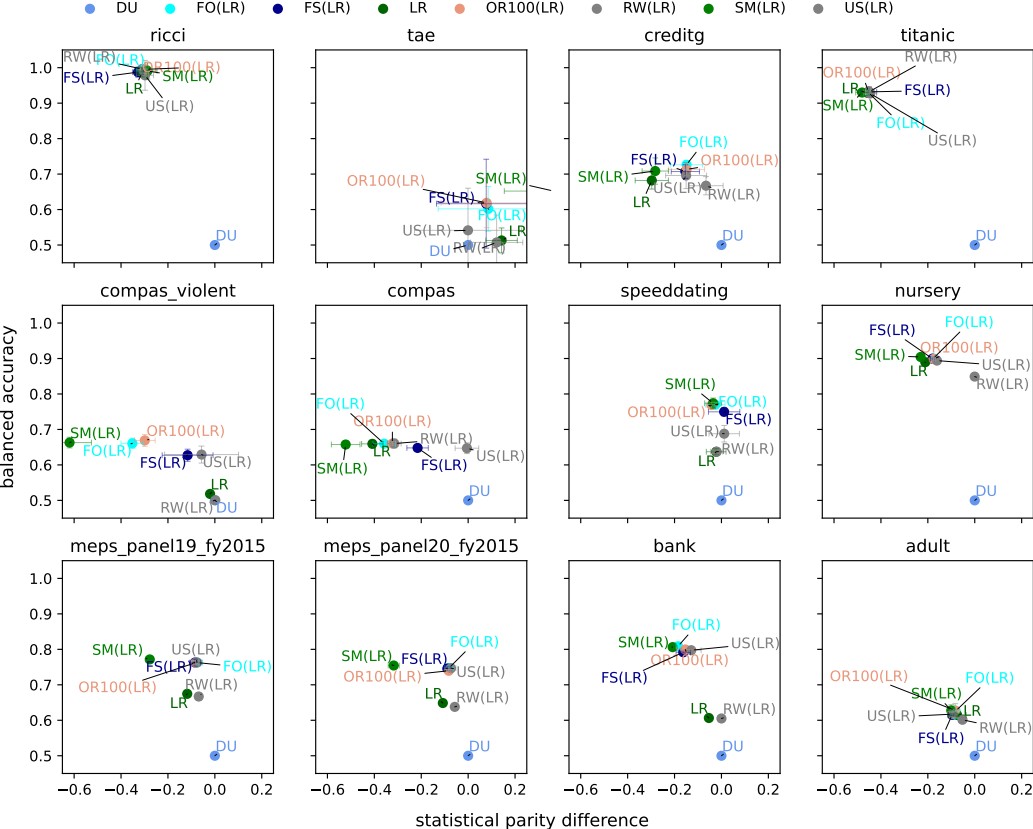

Figure 7: Comparing class imbalance mitigators using statistical parity difference (c.f. Figure 4).

- Average odds difference, AOD =
$$\frac{1}{2}\Big(\Pr(\hat{y}{=}1 \wedge y{=}0 \mid g{=}0) - \Pr(\hat{y}{=}1 \wedge y{=}0 \mid g{=}1) + \Pr(\hat{y}{=}1 \wedge y{=}1 \mid g{=}0) - \Pr(\hat{y}{=}1 \wedge y{=}1 \mid g{=}1)\Big).$$
Average odds difference is the mean of the difference of the false positive rate for the unprivileged and privileged group and the difference of the true positive rate for the unprivileged and privileged group. It is in $[-1, 1]$ and its ideal value is 0. Figure 9 shows the results. Overall, the qualitative conclusions for average odds difference are similar to those for equal opportunity difference.

Some of the formulas for the fairness metrics given above are calculated from only $\hat{y}$ and $g$, whereas others are calculated from $y$, $\hat{y}$, and $g$. Metrics that only require $\hat{y}$ and $g$ can be either computed using the ground-truth labels from the data or the model predictions. On the other hand, metrics that require $y$, $\hat{y}$, and $g$ need both ground truth labels and model predictions to calculate false positive rates or true positive rates. That makes them less suitable for rebalancing data before training a model, because at that time, there are no model predictions yet.

## F  Results with different repair levels

Figure 1 in the main paper showed results for ORBIS with different repair levels for one dataset only, namely meps20. For completeness, we show the results for all 12 datasets in Figures 10–13.

## G  Tabular form of scatter-plot figures

Tables 2–9 present the same data as the scatter plots earlier in the paper.

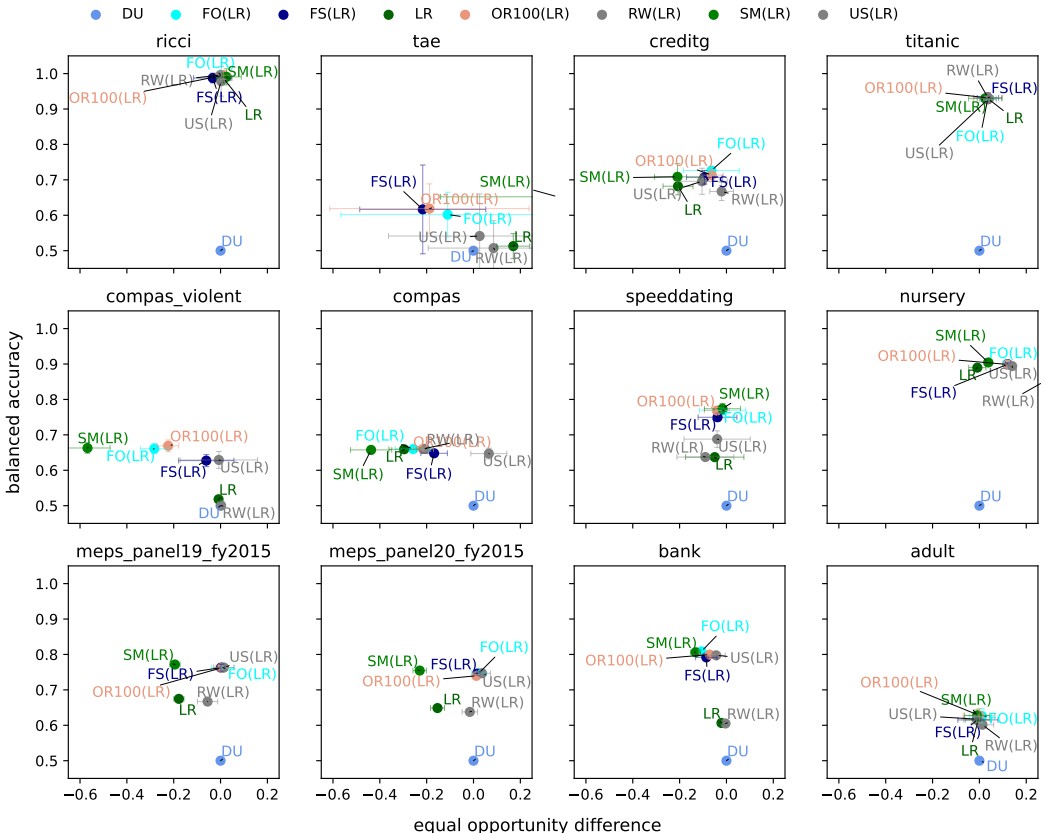

Figure 8: Comparing class imbalance mitigators using equal opportunity difference (c.f. Figure 4).

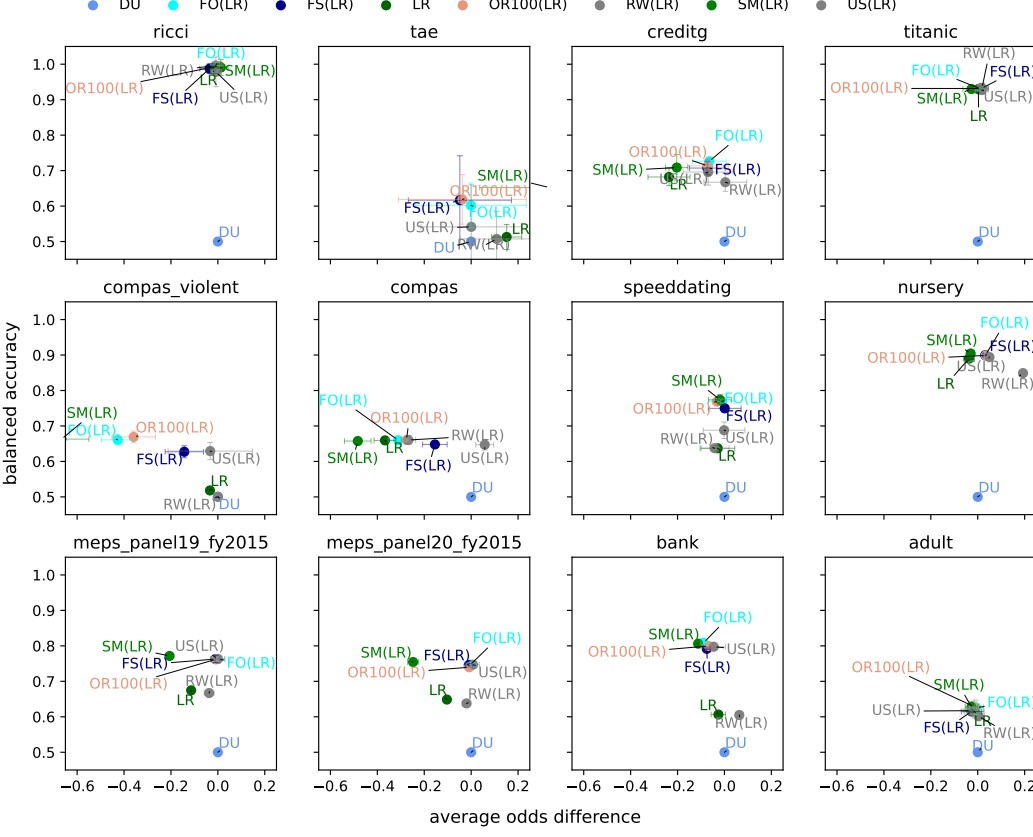

Figure 9: Comparing class imbalance mitigators using average odds difference (c.f. Figure 4).

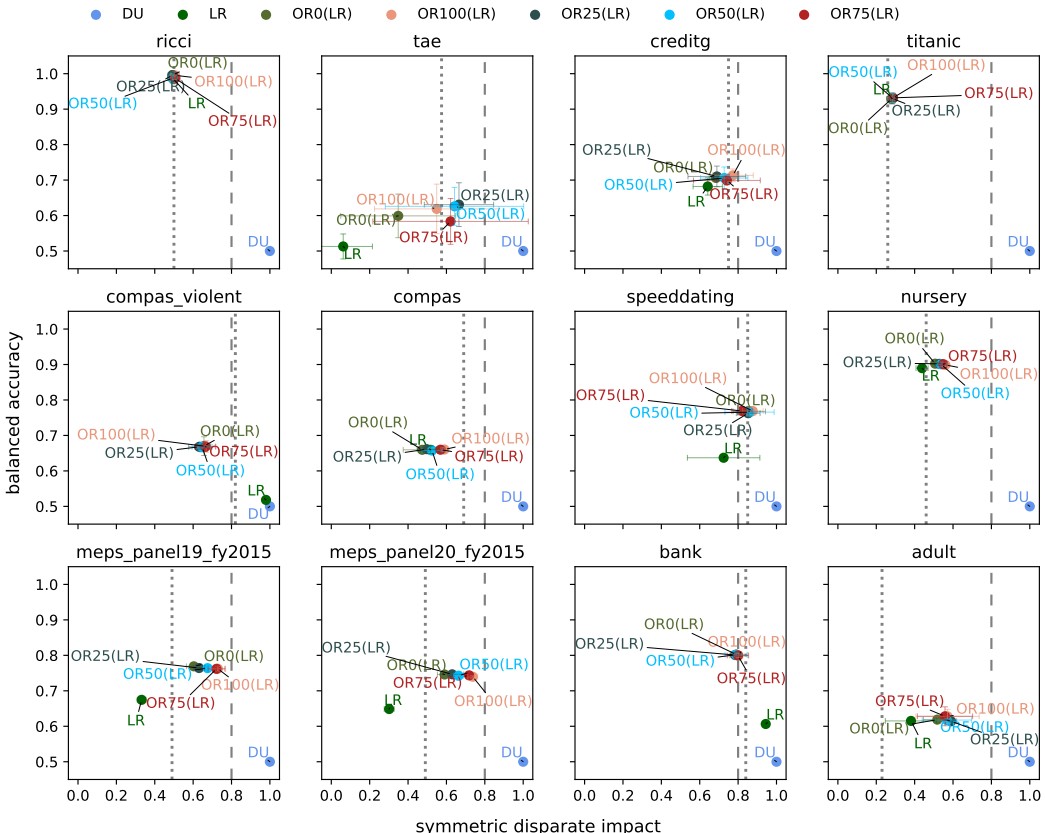

Figure 10: The effect of repair levels with ORBIS and LR (logistic regression).

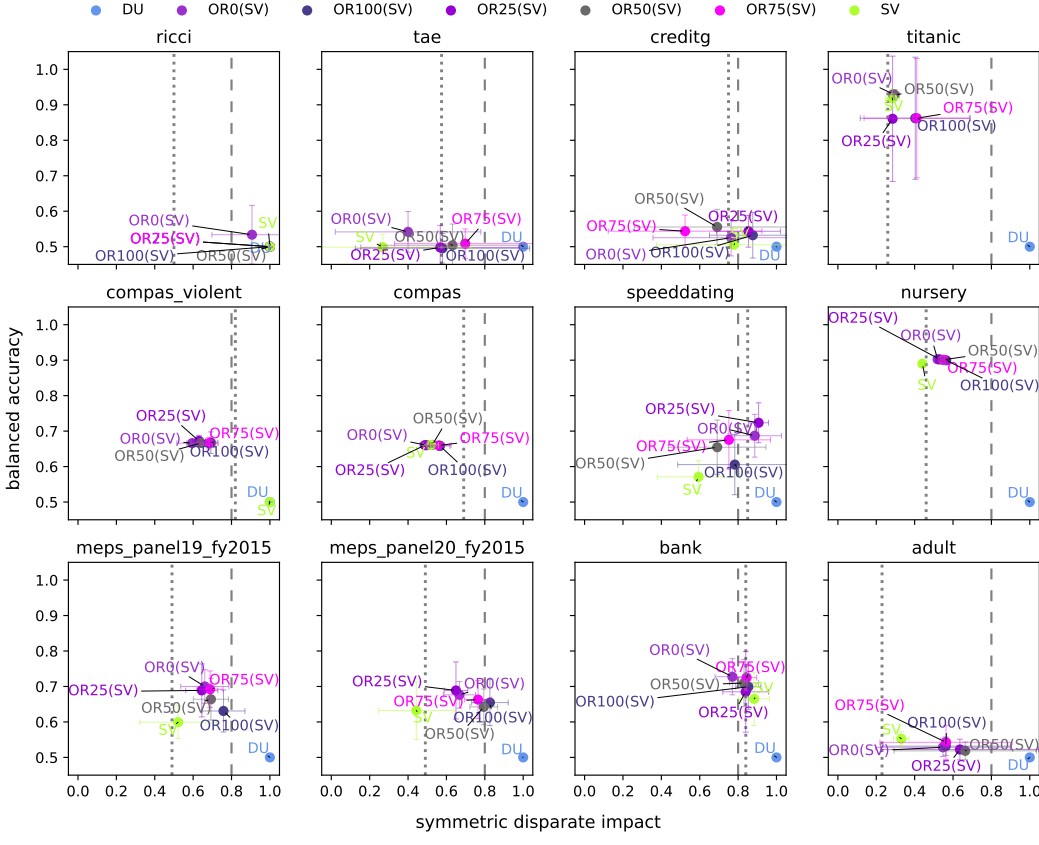

Figure 11: The effect of repair levels with ORBIS and SV (support vector machine).

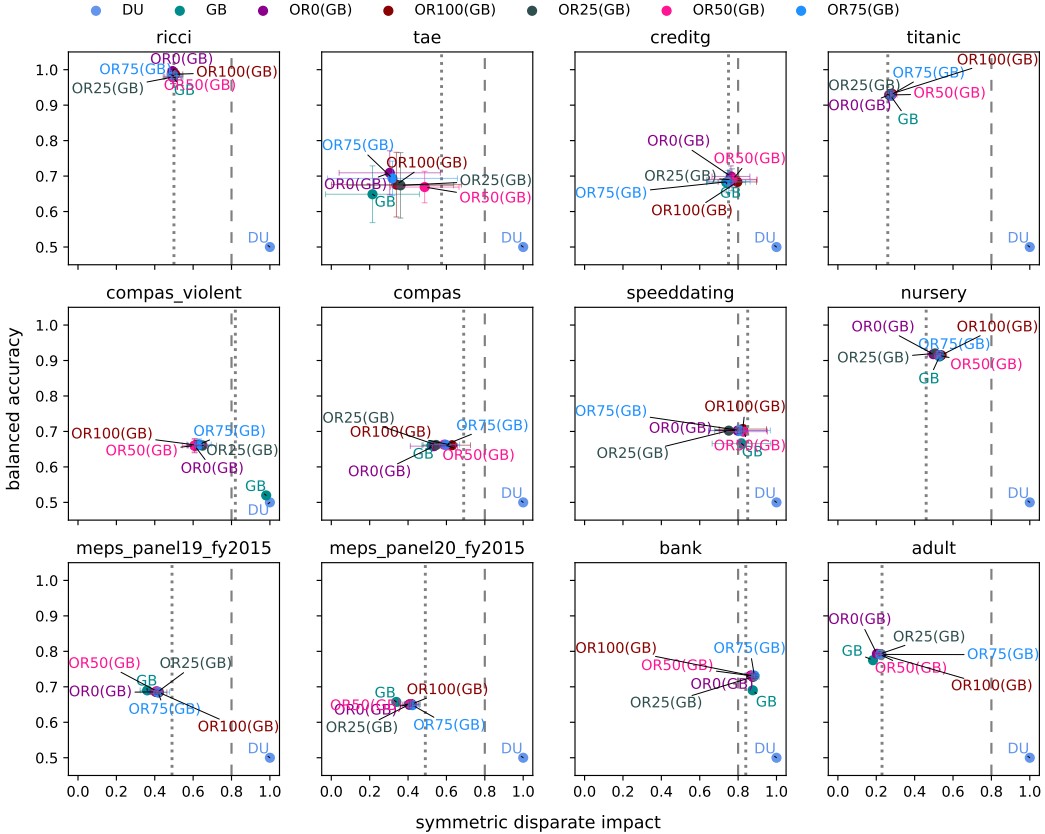

Figure 12: The effect of repair levels with ORBIS and GB (gradient boosting).

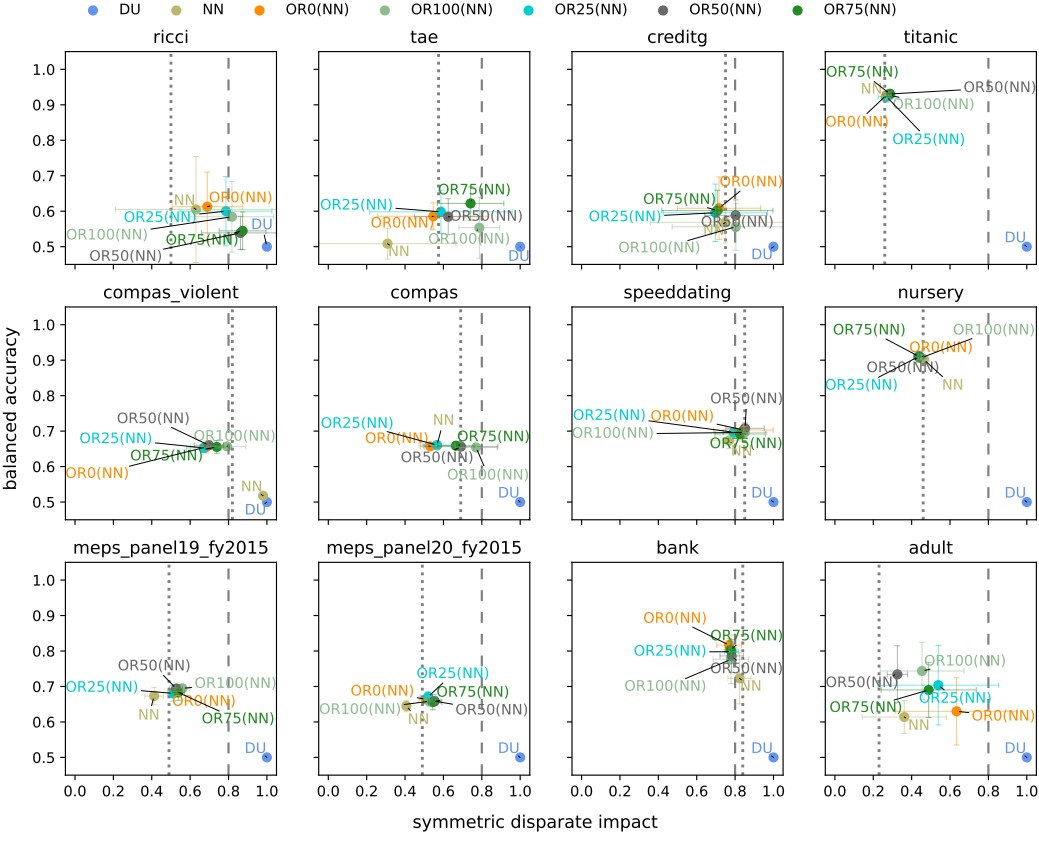

Figure 13: The effect of repair levels with ORBIS and NN (neural network).

Table 2: Tabular form of Figure 1. BA denotes balanced accuracy and DI denotes symmetric disparate impact. The standard deviations are posted in the subscript of the mean value. The scores for the method on Pareto front are in blue. The scores matching the performance of the Dummy Estimator (DU) are in red.

| Method | BA | DI (0.49) | Method | BA | DI (0.49) |
|---|---|---|---|---|---|
| DU | $0.50_{0.00}$ | $1.00_{0.00}$ | DU | $0.50_{0.00}$ | $1.00_{0.00}$ |
| LR | $0.65_{0.01}$ | $0.30_{0.03}$ | SV | $0.63_{0.08}$ | $0.44_{0.20}$ |
| OR0(LR) | $0.75_{0.01}$ | $0.59_{0.04}$ | OR0(SV) | $0.68_{0.04}$ | $0.67_{0.08}$ |
| OR25(LR) | $0.75_{0.01}$ | $0.63_{0.06}$ | OR25(SV) | $0.69_{0.08}$ | $0.65_{0.06}$ |
| OR50(LR) | $0.74_{0.01}$ | $0.66_{0.03}$ | OR50(SV) | $0.64_{0.05}$ | $0.79_{0.07}$ |
| OR75(LR) | $0.74_{0.00}$ | $0.72_{0.02}$ | OR75(SV) | $0.66_{0.05}$ | $0.76_{0.07}$ |
| OR100(LR) | $0.74_{0.00}$ | $0.74_{0.03}$ | OR100(SV) | $0.65_{0.06}$ | $0.83_{0.10}$ |

| Method | BA | DI (0.49) | Method | BA | DI (0.49) |
|---|---|---|---|---|---|
| DU | $0.50_{0.00}$ | $1.00_{0.00}$ | DU | $0.50_{0.00}$ | $1.00_{0.00}$ |
| GB | $0.66_{0.01}$ | $0.34_{0.01}$ | NN | $0.65_{0.02}$ | $0.41_{0.02}$ |
| OR0(GB) | $0.65_{0.01}$ | $0.41_{0.03}$ | OR0(NN) | $0.66_{0.00}$ | $0.51_{0.04}$ |
| OR25(GB) | $0.65_{0.00}$ | $0.41_{0.06}$ | OR25(NN) | $0.67_{0.01}$ | $0.52_{0.04}$ |
| OR50(GB) | $0.65_{0.01}$ | $0.41_{0.03}$ | OR50(NN) | $0.66_{0.01}$ | $0.55_{0.03}$ |
| OR75(GB) | $0.65_{0.01}$ | $0.42_{0.04}$ | OR75(NN) | $0.66_{0.02}$ | $0.54_{0.05}$ |
| OR100(GB) | $0.65_{0.01}$ | $0.42_{0.04}$ | OR100(NN) | $0.66_{0.01}$ | $0.53_{0.03}$ |

Table 3: Tabular view of Figure 4. The presentation follows that of Table 2. $^{\dagger}$ denotes the case where the base disparate impact is greater than 1 and the symmetric disparate impact DI is its reciprocal.

| Method | Ricci BA | DI (0.50) | TAE BA | DI ($1.74^{\dagger}$) | Credit-g BA | DI (0.75) | Titanic BA | DI (0.26) |
|---|---|---|---|---|---|---|---|---|
| DU | $0.50_{0.00}$ | $1.00_{0.00}$ | $0.50_{0.00}$ | $1.00_{0.00}$ | $0.50_{0.00}$ | $1.00_{0.00}$ | $0.50_{0.00}$ | $1.00_{0.00}$ |
| FO(LR) | $1.00_{0.01}$ | $0.49_{0.02}$ | $0.60_{0.06}$ | $0.62_{0.18}$ | $0.73_{0.01}$ | $0.76_{0.11}$ | $0.93_{0.01}$ | $0.29_{0.01}$ |
| FS(LR) | $0.99_{0.01}$ | $0.47_{0.03}$ | $0.62_{0.13}$ | $0.68_{0.26}$ | $0.71_{0.02}$ | $0.75_{0.09}$ | $0.93_{0.01}$ | $0.29_{0.02}$ |
| LR | $0.99_{0.01}$ | $0.49_{0.03}$ | $0.51_{0.04}$ | $0.06_{0.15}$ | $0.68_{0.02}$ | $0.64_{0.08}$ | $0.93_{0.01}$ | $0.28_{0.01}$ |
| OR100(LR) | $1.00_{0.01}$ | $0.51_{0.02}$ | $0.62_{0.07}$ | $0.55_{0.32}$ | $0.71_{0.02}$ | $0.77_{0.11}$ | $0.93_{0.01}$ | $0.29_{0.01}$ |
| RW(LR) | $1.00_{0.01}$ | $0.49_{0.02}$ | $0.51_{0.08}$ | $0.19_{0.31}$ | $0.67_{0.03}$ | $0.90_{0.07}$ | $0.93_{0.01}$ | $0.29_{0.02}$ |
| SM(LR) | $0.99_{0.02}$ | $0.51_{0.04}$ | $0.65_{0.05}$ | $0.34_{0.35}$ | $0.71_{0.04}$ | $0.58_{0.08}$ | $0.93_{0.01}$ | $0.27_{0.02}$ |
| US(LR) | $0.98_{0.04}$ | $0.49_{0.02}$ | $0.54_{0.12}$ | $0.68_{0.21}$ | $0.70_{0.04}$ | $0.75_{0.14}$ | $0.93_{0.01}$ | $0.30_{0.02}$ |

| Method | Compas Violent BA | DI (0.82) | Compas BA | DI (0.69) | Speed Dating BA | DI (0.85) | Nursery BA | DI (0.46) |
|---|---|---|---|---|---|---|---|---|
| DU | $0.50_{0.00}$ | $1.00_{0.00}$ | $0.50_{0.00}$ | $1.00_{0.00}$ | $0.50_{0.00}$ | $1.00_{0.00}$ | $0.50_{0.00}$ | $1.00_{0.00}$ |
| FO(LR) | $0.66_{0.01}$ | $0.58_{0.05}$ | $0.66_{0.00}$ | $0.54_{0.06}$ | $0.77_{0.01}$ | $0.89_{0.07}$ | $0.90_{0.01}$ | $0.57_{0.01}$ |
| FS(LR) | $0.63_{0.02}$ | $0.82_{0.16}$ | $0.65_{0.01}$ | $0.65_{0.08}$ | $0.75_{0.01}$ | $0.83_{0.05}$ | $0.90_{0.00}$ | $0.56_{0.02}$ |
| LR | $0.52_{0.01}$ | $0.98_{0.00}$ | $0.66_{0.01}$ | $0.52_{0.05}$ | $0.64_{0.01}$ | $0.72_{0.19}$ | $0.89_{0.01}$ | $0.44_{0.03}$ |
| OR100(LR) | $0.67_{0.02}$ | $0.67_{0.05}$ | $0.66_{0.01}$ | $0.59_{0.02}$ | $0.77_{0.01}$ | $0.88_{0.05}$ | $0.90_{0.00}$ | $0.56_{0.01}$ |
| RW(LR) | $0.50_{0.00}$ | $1.00_{0.00}$ | $0.66_{0.01}$ | $0.60_{0.03}$ | $0.64_{0.01}$ | $0.79_{0.14}$ | $0.85_{0.01}$ | $0.98_{0.01}$ |
| SM(LR) | $0.66_{0.01}$ | $0.36_{0.09}$ | $0.66_{0.01}$ | $0.37_{0.06}$ | $0.77_{0.01}$ | $0.89_{0.08}$ | $0.90_{0.00}$ | $0.46_{0.02}$ |
| US(LR) | $0.63_{0.02}$ | $0.81_{0.13}$ | $0.65_{0.02}$ | $0.92_{0.04}$ | $0.69_{0.02}$ | $0.87_{0.09}$ | $0.89_{0.00}$ | $0.59_{0.03}$ |

| Method | MEPS19 BA | DI (0.49) | MEPS20 BA | DI (0.49) | Bank BA | DI (0.84) | Adult BA | DI (0.23) |
|---|---|---|---|---|---|---|---|---|
| DU | $0.50_{0.00}$ | $1.00_{0.00}$ | $0.50_{0.00}$ | $1.00_{0.00}$ | $0.50_{0.00}$ | $1.00_{0.00}$ | $0.50_{0.00}$ | $1.00_{0.00}$ |
| FO(LR) | $0.76_{0.01}$ | $0.77_{0.06}$ | $0.75_{0.01}$ | $0.77_{0.04}$ | $0.81_{0.00}$ | $0.75_{0.01}$ | $0.63_{0.02}$ | $0.68_{0.14}$ |
| FS(LR) | $0.76_{0.01}$ | $0.75_{0.06}$ | $0.75_{0.01}$ | $0.75_{0.03}$ | $0.79_{0.00}$ | $0.78_{0.04}$ | $0.62_{0.00}$ | $0.69_{0.08}$ |
| LR | $0.67_{0.00}$ | $0.33_{0.02}$ | $0.65_{0.01}$ | $0.30_{0.03}$ | $0.61_{0.00}$ | $0.94_{0.02}$ | $0.62_{0.00}$ | $0.38_{0.13}$ |
| OR100(LR) | $0.76_{0.01}$ | $0.73_{0.04}$ | $0.74_{0.00}$ | $0.74_{0.03}$ | $0.80_{0.01}$ | $0.80_{0.05}$ | $0.63_{0.02}$ | $0.57_{0.16}$ |
| RW(LR) | $0.67_{0.00}$ | $0.50_{0.04}$ | $0.64_{0.01}$ | $0.51_{0.05}$ | $0.61_{0.00}$ | $0.99_{0.01}$ | $0.60_{0.00}$ | $0.39_{0.08}$ |
| SM(LR) | $0.77_{0.01}$ | $0.40_{0.03}$ | $0.75_{0.01}$ | $0.36_{0.03}$ | $0.81_{0.01}$ | $0.72_{0.02}$ | $0.63_{0.02}$ | $0.70_{0.10}$ |
| US(LR) | $0.76_{0.01}$ | $0.77_{0.05}$ | $0.75_{0.01}$ | $0.79_{0.06}$ | $0.80_{0.00}$ | $0.83_{0.06}$ | $0.62_{0.01}$ | $0.74_{0.08}$ |

Table 4: Tabular view of Figure 5. The presentation follows that of Table 2. The ‡ denotes the case where the estimator predicts the negative outcome for all test points, leading to an undefined DI; we assign it a DI of 1.00 since it is perfectly fair (as inaccurate and fair as the dummy estimator (DU)).

| Method | Ricci BA | DI (0.50) | TAE BA | DI ($1.74^{\dagger}$) | Credit-g BA | DI (0.75) | Titanic BA | DI (0.26) |
|---|---|---|---|---|---|---|---|---|
| AD | $0.50_{0.01}$ | $0.99_{0.03}$ | $0.48_{0.09}$ | $0.79_{0.28}$ | $0.55_{0.07}$ | $0.91_{0.15}$ | $0.77_{0.16}$ | $0.52_{0.37}$ |
| CE(LR) | $1.00_{0.01}$ | $0.48_{0.04}$ | $0.50_{0.03}$ | $0.00_{0.00}$ | $0.57_{0.01}$ | $0.71_{0.03}$ | $0.90_{0.01}$ | $0.22_{0.02}$ |
| DI(LR) | $0.80_{0.05}$ | $0.87_{0.08}$ | $0.52_{0.05}$ | $0.42_{0.27}$ | $0.67_{0.01}$ | $0.84_{0.13}$ | $0.93_{0.01}$ | $0.29_{0.01}$ |
| DU | $0.50_{0.00}$ | $1.00_{0.00}$ | $0.50_{0.00}$ | $1.00_{0.00}$ | $0.50_{0.00}$ | $1.00_{0.00}$ | $0.50_{0.00}$ | $1.00_{0.00}$ |
| EO(LR) | $1.00_{0.00}$ | $0.50_{0.01}$ | $0.52_{0.07}$ | $0.90_{0.09}$ | $0.63_{0.04}$ | $0.88_{0.10}$ | $0.93_{0.01}$ | $0.28_{0.02}$ |
| GF | $0.91_{0.06}$ | $0.51_{0.11}$ | $0.50_{0.00}$ | $1.00_{0.00}$ | $0.69_{0.03}$ | $0.73_{0.12}$ | $0.93_{0.01}$ | $0.28_{0.01}$ |
| LF(LR) | $0.91_{0.20}$ | $0.49_{0.02}$ | $0.58_{0.06}$ | $0.34_{0.39}$ | $0.50_{0.01}$ | $0.99_{0.02}$ | $0.59_{0.04}$ | $0.50_{0.08}$ |
| LR | $0.99_{0.01}$ | $0.49_{0.03}$ | $0.51_{0.04}$ | $0.06_{0.15}$ | $0.68_{0.02}$ | $0.64_{0.08}$ | $0.93_{0.01}$ | $0.28_{0.01}$ |
| MF | $0.77_{0.07}$ | $0.56_{0.05}$ | $0.58_{0.08}$ | $0.49_{0.29}$ | $0.65_{0.03}$ | $0.78_{0.15}$ | $0.85_{0.20}$ | $0.38_{0.28}$ |
| OR100(LR) | $1.00_{0.01}$ | $0.51_{0.02}$ | $0.62_{0.07}$ | $0.55_{0.32}$ | $0.71_{0.02}$ | $0.77_{0.11}$ | $0.93_{0.01}$ | $0.29_{0.01}$ |
| PR | $0.72_{0.03}$ | $0.04_{0.06}$ | $0.51_{0.03}$ | $0.24_{0.32}$ | $0.66_{0.03}$ | $0.79_{0.14}$ | $0.93_{0.00}$ | $0.28_{0.02}$ |
| RO(LR) | $0.97_{0.01}$ | $0.50_{0.08}$ | $0.63_{0.07}$ | $0.44_{0.15}$ | $0.72_{0.02}$ | $0.79_{0.14}$ | $0.93_{0.02}$ | $0.29_{0.01}$ |

| Method | Compas Violent BA | DI (0.82) | Compas BA | DI (0.69) | Speed Dating BA | DI (0.85) | Nursery BA | DI (0.46) |
|---|---|---|---|---|---|---|---|---|
| AD | $0.59_{0.07}$ | $0.81_{0.17}$ | $0.65_{0.01}$ | $0.40_{0.09}$ | $0.65_{0.02}$ | $0.71_{0.24}$ | $0.85_{0.11}$ | $0.56_{0.25}$ |
| CE(LR) | $0.48_{0.01}$ | $0.28_{0.44}$ | $0.37_{0.00}$ | $0.41_{0.03}$ | $0.57_{0.01}$ | $0.58_{0.18}$ | $0.85_{0.01}$ | $0.99_{0.01}$ |
| DI(LR) | $0.52_{0.01}$ | $0.99_{0.01}$ | $0.66_{0.01}$ | $0.59_{0.03}$ | $0.65_{0.03}$ | $0.58_{0.26}$ | $0.85_{0.00}$ | $0.98_{0.01}$ |
| DU | $0.50_{0.00}$ | $1.00_{0.00}$ | $0.50_{0.00}$ | $1.00_{0.00}$ | $0.50_{0.00}$ | $1.00_{0.00}$ | $0.50_{0.00}$ | $1.00_{0.00}$ |
| EO(LR) | $0.46_{0.02}$ | $0.98_{0.01}$ | $0.51_{0.01}$ | $0.93_{0.05}$ | $0.58_{0.01}$ | $0.84_{0.10}$ | $0.82_{0.01}$ | $0.64_{0.02}$ |
| GF | $0.50_{0.00}$ | $1.00_{0.00}$ | $0.50_{0.00}$ | $1.00_{0.00}$ | $0.56_{0.00}$ | $0.52_{0.11}$ | $0.88_{0.00}$ | $0.49_{0.02}$ |
| LF(LR) | $0.52_{0.01}$ | $0.98_{0.01}$ | $0.66_{0.01}$ | $0.49_{0.10}$ | $0.51_{0.02}$ | $0.35_{nan}$ | $0.88_{0.00}$ | $0.49_{0.03}$ |
| LR | $0.52_{0.01}$ | $0.98_{0.00}$ | $0.66_{0.01}$ | $0.52_{0.05}$ | $0.64_{0.01}$ | $0.72_{0.19}$ | $0.89_{0.01}$ | $0.44_{0.03}$ |
| MF | $0.51_{0.01}$ | $0.99_{0.01}$ | $0.66_{0.02}$ | $0.58_{0.03}$ | $0.75_{0.02}$ | $0.92_{0.06}$ | $0.77_{0.04}$ | $0.79_{0.17}$ |
| OR100(LR) | $0.67_{0.02}$ | $0.67_{0.05}$ | $0.66_{0.01}$ | $0.59_{0.02}$ | $0.77_{0.01}$ | $0.88_{0.05}$ | $0.90_{0.00}$ | $0.56_{0.01}$ |
| PR | $0.52_{0.01}$ | $0.98_{0.00}$ | $0.66_{0.01}$ | $0.49_{0.04}$ | $0.64_{0.01}$ | $0.65_{0.16}$ | $0.91_{0.00}$ | $0.45_{0.01}$ |
| RO(LR) | $0.50_{0.00}$ | $1.00_{0.00}$ | $0.50_{0.00}$ | $1.00_{0.00}$ | $0.77_{0.01}$ | $0.85_{0.09}$ | $0.86_{0.00}$ | $0.97_{0.02}$ |

| Method | MEPS19 BA | DI (0.49) | MEPS20 BA | DI (0.49) | Bank BA | DI (0.84) | Adult BA | DI (0.23) |
|---|---|---|---|---|---|---|---|---|
| AD | $0.67_{0.01}$ | $0.63_{0.09}$ | $0.64_{0.02}$ | $0.68_{0.07}$ | $0.64_{0.05}$ | $0.90_{0.03}$ | | |
| CE(LR) | $0.61_{0.00}$ | $0.00_{0.00}$ | $0.59_{0.01}$ | $0.00_{0.00}$ | $0.50_{0.00}$ | $0.95_{0.01}$ | $0.59_{0.00}$ | $0.42_{0.02}$ |
| DI(LR) | $0.67_{0.01}$ | $0.44_{0.02}$ | $0.65_{0.01}$ | $0.44_{0.02}$ | $0.63_{0.02}$ | $0.75_{0.03}$ | $0.61_{0.01}$ | $0.40_{0.05}$ |
| DU | $0.50_{0.00}$ | $1.00_{0.00}$ | $0.50_{0.00}$ | $1.00_{0.00}$ | $0.50_{0.00}$ | $1.00_{0.00}$ | $0.50_{0.00}$ | $1.00_{0.00}$ |
| EO(LR) | $0.65_{0.00}$ | $0.67_{0.06}$ | $0.63_{0.01}$ | $0.71_{0.10}$ | $0.56_{0.03}$ | $0.98_{0.02}$ | $0.59_{0.00}$ | $0.49_{0.10}$ |
| GF | $0.66_{0.00}$ | $0.39_{0.05}$ | $0.64_{0.01}$ | $0.37_{0.02}$ | $0.63_{0.01}$ | $0.90_{0.01}$ | $0.64_{0.04}$ | $0.14_{0.03}$ |
| LF(LR) | $0.50_{0.00}$ | $0.37_{0.16}$ | $0.50_{0.00}$ | $0.62_{nan}$ | $0.50_{0.00}$ | $1.00_{0.00}$ | $0.51_{0.01}$ | $0.46_{nan}$ |
| LR | $0.67_{0.00}$ | $0.33_{0.02}$ | $0.65_{0.01}$ | $0.30_{0.03}$ | $0.61_{0.00}$ | $0.94_{0.02}$ | $0.62_{0.00}$ | $0.38_{0.13}$ |
| MF | $0.77_{0.01}$ | $0.56_{0.12}$ | $0.76_{0.01}$ | $0.53_{0.11}$ | $0.63_{0.01}$ | $0.92_{0.01}$ | $0.38_{0.01}$ | $0.96_{0.03}$ |
| OR100(LR) | $0.76_{0.01}$ | $0.73_{0.04}$ | $0.74_{0.00}$ | $0.74_{0.03}$ | $0.80_{0.01}$ | $0.80_{0.05}$ | $0.63_{0.02}$ | $0.57_{0.16}$ |
| PR | $0.67_{0.01}$ | $0.36_{0.05}$ | $0.64_{0.01}$ | $0.38_{0.03}$ | $0.66_{0.00}$ | $0.86_{0.03}$ | $0.62_{0.00}$ | $0.14_{0.02}$ |
| RO(LR) | $0.77_{0.01}$ | $0.55_{0.03}$ | $0.76_{0.01}$ | $0.57_{0.02}$ | $0.80_{0.01}$ | $0.85_{0.07}$ | $0.62_{0.00}$ | $0.43_{0.08}$ |

Table 5: Tabular view of Figure 6. The presentation follows that of Table 2.

| Method | Ricci BA | DI (0.50) | TAE BA | DI (1.74†) | Credit-g BA | DI (0.75) | Titanic BA | DI (0.26) |
|---|---|---|---|---|---|---|---|---|
| A_AM | $0.66_{0.30}$ | $0.83_{0.26}$ | $0.54_{0.07}$ | $0.81_{nan}$ | $0.63_{0.08}$ | $0.70_{0.14}$ | $0.67_{0.23}$ | $0.73_{0.38}$ |
| A_GM | $0.50_{0.00}$ | $0.87_{0.16}$ | $0.53_{0.10}$ | $0.48_{0.50}$ | $0.65_{0.09}$ | $0.94_{0.03}$ | $0.93_{0.01}$ | $0.29_{0.01}$ |
| A_HM | $0.54_{0.07}$ | $0.97_{0.05}$ | $0.52_{0.06}$ | $0.84_{0.14}$ | $0.69_{0.01}$ | $0.67_{0.18}$ | $0.93_{0.01}$ | $0.29_{0.02}$ |
| A_HT | $0.56_{0.13}$ | $0.89_{0.12}$ | $0.66_{0.05}$ | $0.40_{0.04}$ | $0.51_{0.01}$ | $0.66_{0.57}$ | $0.93_{0.01}$ | $0.28_{0.03}$ |
| A_ST | $0.63_{0.10}$ | $0.72_{0.20}$ | $0.58_{0.08}$ | $0.69_{0.28}$ | $0.72_{0.02}$ | $0.71_{0.04}$ | $0.93_{0.02}$ | $0.29_{0.01}$ |

| Method | Compas Violent BA | DI (0.82) | Compas BA | DI (0.69) | Speed Dating BA | DI (0.85) | Nursery BA | DI (0.46) |
|---|---|---|---|---|---|---|---|---|
| A_AM | $0.52_{0.01}$ | $0.98_{0.00}$ | $0.65_{0.01}$ | $0.64_{0.06}$ | $0.75_{0.01}$ | $0.83_{0.10}$ | $0.90_{0.00}$ | $0.56_{0.02}$ |
| A_GM | $0.52_{0.01}$ | $0.97_{0.01}$ | $0.66_{0.01}$ | $0.68_{0.13}$ | $0.70_{0.10}$ | $0.80_{0.14}$ | $0.90_{0.01}$ | $0.56_{0.00}$ |
| A_HM | $0.63_{0.03}$ | $0.78_{0.05}$ | $0.66_{0.02}$ | $0.68_{0.16}$ | $0.76_{0.01}$ | $0.92_{0.04}$ | $0.90_{0.01}$ | $0.55_{0.02}$ |
| A_HT | $0.53_{0.01}$ | $0.97_{0.01}$ | $0.66_{0.01}$ | $0.70_{0.06}$ | $0.69_{0.09}$ | $0.76_{0.10}$ | $0.90_{0.01}$ | $0.55_{0.01}$ |
| A_ST | $0.57_{0.07}$ | $0.87_{0.17}$ | $0.66_{0.01}$ | $0.67_{0.18}$ | $0.77_{0.01}$ | $0.82_{0.05}$ | $0.90_{0.01}$ | $0.54_{0.01}$ |

| Method | MEPS19 BA | DI (0.49) | MEPS20 BA | DI (0.49) | Bank BA | DI (0.84) | Adult BA | DI (0.23) |
|---|---|---|---|---|---|---|---|---|
| A_AM | $0.76_{0.02}$ | $0.68_{0.03}$ | $0.68_{0.04}$ | $0.84_{0.08}$ | $0.77_{0.04}$ | $0.84_{0.03}$ | $0.55_{0.06}$ | $0.76_{0.40}$ |
| A_GM | $0.76_{0.01}$ | $0.70_{0.05}$ | $0.72_{0.05}$ | $0.72_{0.02}$ | $0.78_{0.04}$ | $0.78_{0.12}$ | $0.67_{0.09}$ | $0.61_{0.23}$ |
| A_HM | $0.76_{0.01}$ | $0.73_{0.02}$ | $0.69_{0.05}$ | $0.76_{0.07}$ | $0.75_{0.05}$ | $0.87_{0.06}$ | $0.55_{0.03}$ | $0.51_{0.39}$ |
| A_HT | $0.71_{0.10}$ | $0.69_{0.07}$ | $0.70_{0.05}$ | $0.71_{0.07}$ | $0.73_{0.04}$ | $0.84_{0.04}$ | $0.56_{0.06}$ | $0.58_{0.37}$ |
| A_ST | $0.74_{0.01}$ | $0.69_{0.08}$ | $0.66_{0.02}$ | $0.70_{0.10}$ | $0.80_{0.01}$ | $0.74_{0.02}$ | $0.54_{0.03}$ | $0.49_{0.44}$ |

Table 6: Tabular view of Figure 10. The presentation follows that of Table 2.

| Method | Ricci BA | DI (0.50) | TAE BA | DI (1.74†) | Credit-g BA | DI (0.75) | Titanic BA | DI (0.26) |
|---|---|---|---|---|---|---|---|---|
| DU | $0.50_{0.00}$ | $1.00_{0.00}$ | $0.50_{0.00}$ | $1.00_{0.00}$ | $0.50_{0.00}$ | $1.00_{0.00}$ | $0.50_{0.00}$ | $1.00_{0.00}$ |
| LR | $0.99_{0.01}$ | $0.49_{0.03}$ | $0.51_{0.04}$ | $0.06_{0.15}$ | $0.68_{0.02}$ | $0.64_{0.08}$ | $0.93_{0.01}$ | $0.28_{0.01}$ |
| OR0(LR) | $1.00_{0.01}$ | $0.49_{0.02}$ | $0.60_{0.06}$ | $0.35_{0.30}$ | $0.71_{0.01}$ | $0.68_{0.04}$ | $0.93_{0.01}$ | $0.28_{0.03}$ |
| OR100(LR) | $1.00_{0.01}$ | $0.51_{0.02}$ | $0.62_{0.07}$ | $0.55_{0.32}$ | $0.71_{0.02}$ | $0.77_{0.11}$ | $0.93_{0.01}$ | $0.29_{0.01}$ |
| OR25(LR) | $1.00_{0.01}$ | $0.49_{0.02}$ | $0.63_{0.06}$ | $0.67_{0.18}$ | $0.71_{0.03}$ | $0.69_{0.15}$ | $0.93_{0.02}$ | $0.28_{0.02}$ |
| OR50(LR) | $0.99_{0.01}$ | $0.49_{0.03}$ | $0.63_{0.05}$ | $0.64_{0.36}$ | $0.71_{0.03}$ | $0.73_{0.12}$ | $0.93_{0.01}$ | $0.29_{0.01}$ |
| OR75(LR) | $0.99_{0.01}$ | $0.51_{0.03}$ | $0.58_{0.06}$ | $0.62_{0.41}$ | $0.70_{0.01}$ | $0.74_{0.17}$ | $0.93_{0.00}$ | $0.29_{0.01}$ |

| Method | Compas Violent BA | DI (0.82) | Compas BA | DI (0.69) | Speed Dating BA | DI (0.85) | Nursery BA | DI (0.46) |
|---|---|---|---|---|---|---|---|---|
| DU | $0.50_{0.00}$ | $1.00_{0.00}$ | $0.50_{0.00}$ | $1.00_{0.00}$ | $0.50_{0.00}$ | $1.00_{0.00}$ | $0.50_{0.00}$ | $1.00_{0.00}$ |
| LR | $0.52_{0.01}$ | $0.98_{0.00}$ | $0.66_{0.01}$ | $0.52_{0.05}$ | $0.64_{0.01}$ | $0.72_{0.19}$ | $0.89_{0.01}$ | $0.44_{0.03}$ |
| OR0(LR) | $0.67_{0.03}$ | $0.66_{0.04}$ | $0.66_{0.01}$ | $0.47_{0.10}$ | $0.77_{0.01}$ | $0.86_{0.08}$ | $0.90_{0.00}$ | $0.51_{0.01}$ |
| OR100(LR) | $0.67_{0.02}$ | $0.67_{0.05}$ | $0.66_{0.01}$ | $0.59_{0.02}$ | $0.77_{0.01}$ | $0.88_{0.05}$ | $0.90_{0.00}$ | $0.56_{0.01}$ |
| OR25(LR) | $0.67_{0.01}$ | $0.63_{0.05}$ | $0.66_{0.01}$ | $0.50_{0.05}$ | $0.76_{0.01}$ | $0.85_{0.06}$ | $0.90_{0.00}$ | $0.53_{0.02}$ |
| OR50(LR) | $0.67_{0.01}$ | $0.65_{0.05}$ | $0.66_{0.01}$ | $0.52_{0.05}$ | $0.77_{0.01}$ | $0.86_{0.13}$ | $0.90_{0.01}$ | $0.54_{0.03}$ |
| OR75(LR) | $0.67_{0.01}$ | $0.66_{0.05}$ | $0.66_{0.01}$ | $0.57_{0.02}$ | $0.77_{0.01}$ | $0.82_{0.05}$ | $0.90_{0.00}$ | $0.55_{0.02}$ |

| Method | MEPS19 BA | DI (0.49) | MEPS20 BA | DI (0.49) | Bank BA | DI (0.84) | Adult BA | DI (0.23) |
|---|---|---|---|---|---|---|---|---|
| DU | $0.50_{0.00}$ | $1.00_{0.00}$ | $0.50_{0.00}$ | $1.00_{0.00}$ | $0.50_{0.00}$ | $1.00_{0.00}$ | $0.50_{0.00}$ | $1.00_{0.00}$ |
| LR | $0.67_{0.00}$ | $0.33_{0.02}$ | $0.65_{0.01}$ | $0.30_{0.03}$ | $0.61_{0.00}$ | $0.94_{0.02}$ | $0.62_{0.00}$ | $0.38_{0.13}$ |
| OR0(LR) | $0.77_{0.01}$ | $0.60_{0.04}$ | $0.75_{0.01}$ | $0.59_{0.04}$ | $0.80_{0.01}$ | $0.79_{0.03}$ | $0.62_{0.01}$ | $0.52_{0.07}$ |
| OR100(LR) | $0.76_{0.01}$ | $0.73_{0.04}$ | $0.74_{0.00}$ | $0.74_{0.03}$ | $0.80_{0.01}$ | $0.80_{0.05}$ | $0.63_{0.02}$ | $0.57_{0.16}$ |
| OR25(LR) | $0.76_{0.01}$ | $0.63_{0.03}$ | $0.75_{0.01}$ | $0.63_{0.06}$ | $0.80_{0.00}$ | $0.78_{0.03}$ | $0.61_{0.00}$ | $0.59_{0.03}$ |
| OR50(LR) | $0.76_{0.01}$ | $0.68_{0.02}$ | $0.74_{0.01}$ | $0.66_{0.03}$ | $0.80_{0.00}$ | $0.79_{0.06}$ | $0.62_{0.00}$ | $0.57_{0.13}$ |
| OR75(LR) | $0.76_{0.01}$ | $0.72_{0.04}$ | $0.74_{0.00}$ | $0.72_{0.02}$ | $0.80_{0.00}$ | $0.80_{0.06}$ | $0.63_{0.03}$ | $0.56_{0.14}$ |

Table 7: Tabular view of Figure 11. The presentation follows that of Table 2.

| Method | Ricci BA | DI (0.50) | TAE BA | DI ($1.74^{\dagger}$) | Credit-g BA | DI (0.75) | Titanic BA | DI (0.26) |
|---|---|---|---|---|---|---|---|---|
| DU | $0.50_{0.00}$ | $1.00_{0.00}$ | $0.50_{0.00}$ | $1.00_{0.00}$ | $0.50_{0.00}$ | $1.00_{0.00}$ | $0.50_{0.00}$ | $1.00_{0.00}$ |
| SV | $0.50_{0.00}$ | $1.00_{nan}$ | $0.50_{0.04}$ | $0.27_{0.46}$ | $0.51_{0.01}$ | $0.78_{0.40}$ | $0.92_{0.03}$ | $0.28_{0.04}$ |
| OR0(SV) | $0.53_{0.08}$ | $0.91_{0.21}$ | $0.54_{0.06}$ | $0.40_{0.38}$ | $0.53_{0.05}$ | $0.76_{0.41}$ | $0.93_{0.01}$ | $0.29_{0.02}$ |
| OR100(SV) | $0.50_{0.00}$ | $1.00_{0.00}$ | $0.50_{0.04}$ | $0.58_{0.45}$ | $0.53_{0.06}$ | $0.88_{0.23}$ | $0.86_{0.17}$ | $0.40_{0.29}$ |
| OR25(SV) | $0.50_{0.00}$ | $1.00_{0.00}$ | $0.50_{0.06}$ | $0.57_{0.41}$ | $0.54_{0.04}$ | $0.85_{0.17}$ | $0.86_{0.18}$ | $0.29_{0.01}$ |
| OR50(SV) | $0.50_{0.00}$ | $1.00_{0.00}$ | $0.50_{0.06}$ | $0.63_{0.38}$ | $0.56_{0.05}$ | $0.69_{0.43}$ | $0.93_{0.01}$ | $0.29_{0.03}$ |
| OR75(SV) | $0.50_{0.00}$ | $1.00_{nan}$ | $0.51_{0.04}$ | $0.70_{0.37}$ | $0.54_{0.05}$ | $0.52_{0.40}$ | $0.86_{0.17}$ | $0.41_{0.27}$ |

| Method | Compas BA | Violent DI (0.82) | Compas BA | DI (0.69) | Speed BA | Dating DI (0.85) | Nursery BA | DI (0.46) |
|---|---|---|---|---|---|---|---|---|
| SV | | | | | | | | |
| DU | $0.50_{0.00}$ | $1.00_{0.00}$ | $0.50_{0.00}$ | $1.00_{0.00}$ | $0.50_{0.00}$ | $1.00_{0.00}$ | $0.50_{0.00}$ | $1.00_{0.00}$ |
| SV | $0.50_{0.00}$ | $1.00_{0.00}$ | $0.66_{0.01}$ | $0.52_{0.04}$ | $0.57_{0.05}$ | $0.59_{0.22}$ | $0.89_{0.00}$ | $0.44_{0.02}$ |
| OR0(SV) | $0.67_{0.01}$ | $0.59_{0.08}$ | $0.66_{0.01}$ | $0.50_{0.03}$ | $0.69_{0.06}$ | $0.89_{0.14}$ | $0.90_{0.00}$ | $0.52_{0.02}$ |
| OR100(SV) | $0.67_{0.03}$ | $0.69_{0.03}$ | $0.66_{0.01}$ | $0.57_{0.05}$ | $0.61_{0.08}$ | $0.78_{0.30}$ | $0.90_{0.01}$ | $0.56_{0.01}$ |
| OR25(SV) | $0.67_{0.02}$ | $0.63_{0.03}$ | $0.66_{0.01}$ | $0.49_{0.02}$ | $0.72_{0.06}$ | $0.91_{0.05}$ | $0.90_{0.00}$ | $0.53_{0.02}$ |
| OR50(SV) | $0.67_{0.01}$ | $0.64_{0.07}$ | $0.66_{0.01}$ | $0.53_{0.06}$ | $0.65_{0.08}$ | $0.69_{0.25}$ | $0.90_{0.01}$ | $0.55_{0.01}$ |
| OR75(SV) | $0.67_{0.02}$ | $0.68_{0.05}$ | $0.66_{0.00}$ | $0.56_{0.06}$ | $0.68_{0.08}$ | $0.75_{0.22}$ | $0.90_{0.00}$ | $0.55_{0.02}$ |

| Method | MEPS19 BA | DI (0.49) | MEPS20 BA | DI (0.49) | Bank BA | DI (0.84) | Adult BA | DI (0.23) |
|---|---|---|---|---|---|---|---|---|
| DU | $0.50_{0.00}$ | $1.00_{0.00}$ | $0.50_{0.00}$ | $1.00_{0.00}$ | $0.50_{0.00}$ | $1.00_{0.00}$ | $0.50_{0.00}$ | $1.00_{0.00}$ |
| SV | $0.60_{0.05}$ | $0.52_{0.20}$ | $0.63_{0.08}$ | $0.44_{0.20}$ | $0.67_{0.07}$ | $0.88_{0.08}$ | $0.55_{0.02}$ | $0.33_{0.04}$ |
| OR0(SV) | $0.70_{0.05}$ | $0.66_{0.13}$ | $0.68_{0.04}$ | $0.67_{0.08}$ | $0.73_{0.05}$ | $0.77_{0.09}$ | $0.53_{0.03}$ | $0.55_{0.35}$ |
| OR100(SV) | $0.63_{0.06}$ | $0.76_{0.11}$ | $0.65_{0.06}$ | $0.83_{0.10}$ | $0.70_{0.06}$ | $0.85_{0.07}$ | $0.53_{0.03}$ | $0.56_{0.34}$ |
| OR25(SV) | $0.69_{0.07}$ | $0.65_{0.08}$ | $0.69_{0.08}$ | $0.65_{0.06}$ | $0.68_{0.11}$ | $0.84_{0.12}$ | $0.52_{0.03}$ | $0.64_{0.41}$ |
| OR50(SV) | $0.66_{0.06}$ | $0.69_{0.09}$ | $0.64_{0.05}$ | $0.79_{0.07}$ | $0.71_{0.07}$ | $0.83_{0.08}$ | $0.52_{0.02}$ | $0.66_{0.37}$ |
| OR75(SV) | $0.69_{0.05}$ | $0.69_{0.07}$ | $0.66_{0.05}$ | $0.76_{0.07}$ | $0.72_{0.04}$ | $0.84_{0.05}$ | $0.54_{0.05}$ | $0.56_{0.34}$ |

Table 8: Tabular view of Figure 12. The presentation follows that of Table 2.

| Method | Ricci BA | DI (0.50) | TAE BA | DI ($1.74^{\dagger}$) | Credit-g BA | DI (0.75) | Titanic BA | DI (0.26) |
|---|---|---|---|---|---|---|---|---|
| DU | $0.50_{0.00}$ | $1.00_{0.00}$ | $0.50_{0.00}$ | $1.00_{0.00}$ | $0.50_{0.00}$ | $1.00_{0.00}$ | $0.50_{0.00}$ | $1.00_{0.00}$ |
| GB | $0.98_{0.02}$ | $0.49_{0.04}$ | $0.65_{0.08}$ | $0.21_{0.24}$ | $0.68_{0.03}$ | $0.74_{0.10}$ | $0.93_{0.01}$ | $0.27_{0.01}$ |
| OR0(GB) | $1.00_{0.01}$ | $0.49_{0.02}$ | $0.71_{0.06}$ | $0.30_{0.26}$ | $0.70_{0.03}$ | $0.76_{0.10}$ | $0.93_{0.01}$ | $0.27_{0.02}$ |
| OR100(GB) | $0.99_{0.01}$ | $0.51_{0.04}$ | $0.68_{0.09}$ | $0.34_{0.34}$ | $0.68_{0.01}$ | $0.80_{0.10}$ | $0.93_{0.01}$ | $0.28_{0.02}$ |
| OR25(GB) | $0.98_{0.02}$ | $0.50_{0.05}$ | $0.67_{0.09}$ | $0.36_{0.28}$ | $0.69_{0.03}$ | $0.77_{0.13}$ | $0.93_{0.01}$ | $0.27_{0.02}$ |
| OR50(GB) | $0.98_{0.02}$ | $0.49_{0.05}$ | $0.67_{0.04}$ | $0.49_{0.18}$ | $0.69_{0.02}$ | $0.77_{0.12}$ | $0.93_{0.01}$ | $0.28_{0.03}$ |
| OR75(GB) | $0.98_{0.01}$ | $0.50_{0.04}$ | $0.69_{0.05}$ | $0.32_{0.34}$ | $0.68_{0.03}$ | $0.75_{0.11}$ | $0.93_{0.01}$ | $0.28_{0.01}$ |

| Method | Compas Violent BA | DI (0.82) | Compas BA | DI (0.69) | Speed Dating BA | DI (0.85) | Nursery BA | DI (0.46) |
|---|---|---|---|---|---|---|---|---|
| DU | $0.50_{0.00}$ | $1.00_{0.00}$ | $0.50_{0.00}$ | $1.00_{0.00}$ | $0.50_{0.00}$ | $1.00_{0.00}$ | $0.50_{0.00}$ | $1.00_{0.00}$ |
| GB | $0.52_{0.01}$ | $0.98_{0.01}$ | $0.66_{0.00}$ | $0.52_{0.05}$ | $0.67_{0.01}$ | $0.82_{0.15}$ | $0.91_{0.01}$ | $0.53_{0.02}$ |
| OR0(GB) | $0.66_{0.02}$ | $0.61_{0.04}$ | $0.66_{0.01}$ | $0.53_{0.12}$ | $0.70_{0.01}$ | $0.80_{0.15}$ | $0.92_{0.01}$ | $0.50_{0.03}$ |
| OR100(GB) | $0.66_{0.01}$ | $0.62_{0.05}$ | $0.66_{0.01}$ | $0.63_{0.10}$ | $0.71_{0.01}$ | $0.82_{0.12}$ | $0.91_{0.01}$ | $0.54_{0.03}$ |
| OR25(GB) | $0.66_{0.01}$ | $0.65_{0.07}$ | $0.66_{0.01}$ | $0.55_{0.07}$ | $0.70_{0.01}$ | $0.75_{0.07}$ | $0.92_{0.00}$ | $0.51_{0.02}$ |
| OR50(GB) | $0.66_{0.02}$ | $0.61_{0.06}$ | $0.66_{0.01}$ | $0.59_{0.06}$ | $0.70_{0.01}$ | $0.83_{0.12}$ | $0.91_{0.00}$ | $0.53_{0.01}$ |
| OR75(GB) | $0.66_{0.02}$ | $0.63_{0.06}$ | $0.66_{0.01}$ | $0.60_{0.07}$ | $0.70_{0.01}$ | $0.81_{0.16}$ | $0.91_{0.00}$ | $0.53_{0.02}$ |

| Method | MEPS19 BA | DI (0.49) | MEPS20 BA | DI (0.49) | Bank BA | DI (0.84) | Adult BA | DI (0.23) |
|---|---|---|---|---|---|---|---|---|
| DU | $0.50_{0.00}$ | $1.00_{0.00}$ | $0.50_{0.00}$ | $1.00_{0.00}$ | $0.50_{0.00}$ | $1.00_{0.00}$ | $0.50_{0.00}$ | $1.00_{0.00}$ |
| GB | $0.69_{0.01}$ | $0.36_{0.04}$ | $0.66_{0.01}$ | $0.34_{0.01}$ | $0.69_{0.01}$ | $0.88_{0.02}$ | $0.78_{0.00}$ | $0.18_{0.02}$ |
| OR0(GB) | $0.69_{0.01}$ | $0.41_{0.03}$ | $0.65_{0.01}$ | $0.41_{0.03}$ | $0.73_{0.01}$ | $0.87_{0.02}$ | $0.79_{0.00}$ | $0.20_{0.02}$ |
| OR100(GB) | $0.68_{0.01}$ | $0.42_{0.04}$ | $0.65_{0.01}$ | $0.42_{0.04}$ | $0.73_{0.01}$ | $0.88_{0.02}$ | $0.79_{0.00}$ | $0.23_{0.01}$ |
| OR25(GB) | $0.68_{0.00}$ | $0.41_{0.03}$ | $0.65_{0.00}$ | $0.41_{0.06}$ | $0.73_{0.01}$ | $0.88_{0.03}$ | $0.79_{0.01}$ | $0.22_{0.02}$ |
| OR50(GB) | $0.69_{0.01}$ | $0.41_{0.03}$ | $0.65_{0.01}$ | $0.41_{0.03}$ | $0.73_{0.01}$ | $0.87_{0.02}$ | $0.79_{0.00}$ | $0.22_{0.02}$ |
| OR75(GB) | $0.68_{0.01}$ | $0.42_{0.06}$ | $0.65_{0.01}$ | $0.42_{0.04}$ | $0.73_{0.00}$ | $0.88_{0.02}$ | $0.79_{0.01}$ | $0.22_{0.03}$ |

Table 9: Tabular view of Figure 13. The presentation follows that of Table 2.

| Method | Ricci BA | DI (0.50) | TAE BA | DI ($1.74^{\dagger}$) | Credit-g BA | DI (0.75) | Titanic BA | DI (0.26) |
|---|---|---|---|---|---|---|---|---|
| DU | $0.50_{0.00}$ | $1.00_{0.00}$ | $0.50_{0.00}$ | $1.00_{0.00}$ | $0.50_{0.00}$ | $1.00_{0.00}$ | $0.50_{0.00}$ | $1.00_{0.00}$ |
| NN | $0.60_{0.15}$ | $0.63_{0.42}$ | $0.51_{0.04}$ | $0.31_{0.36}$ | $0.57_{0.06}$ | $0.75_{0.39}$ | $0.92_{0.01}$ | $0.26_{0.01}$ |
| OR0(NN) | $0.61_{0.10}$ | $0.69_{0.18}$ | $0.59_{0.04}$ | $0.55_{0.29}$ | $0.61_{0.09}$ | $0.72_{0.22}$ | $0.93_{0.01}$ | $0.27_{0.02}$ |
| OR100(NN) | $0.58_{0.10}$ | $0.82_{0.22}$ | $0.55_{0.09}$ | $0.79_{0.11}$ | $0.56_{0.07}$ | $0.80_{0.33}$ | $0.93_{0.01}$ | $0.28_{0.03}$ |
| OR25(NN) | $0.60_{0.10}$ | $0.79_{0.24}$ | $0.60_{0.06}$ | $0.59_{0.37}$ | $0.60_{0.08}$ | $0.70_{0.27}$ | $0.92_{0.01}$ | $0.27_{0.04}$ |
| OR50(NN) | $0.54_{0.05}$ | $0.86_{0.20}$ | $0.58_{0.05}$ | $0.63_{0.18}$ | $0.59_{0.04}$ | $0.80_{0.16}$ | $0.93_{0.01}$ | $0.28_{0.01}$ |
| OR75(NN) | $0.54_{0.05}$ | $0.87_{0.12}$ | $0.62_{0.05}$ | $0.74_{0.17}$ | $0.60_{0.06}$ | $0.71_{0.29}$ | $0.93_{0.01}$ | $0.29_{0.01}$ |

| Method | Compas Violent BA | DI (0.82) | Compas BA | DI (0.69) | Speed Dating BA | DI (0.85) | Nursery BA | DI (0.46) |
|---|---|---|---|---|---|---|---|---|
| DU | $0.50_{0.00}$ | $1.00_{0.00}$ | $0.50_{0.00}$ | $1.00_{0.00}$ | $0.50_{0.00}$ | $1.00_{0.00}$ | $0.50_{0.00}$ | $1.00_{0.00}$ |
| NN | $0.52_{0.01}$ | $0.98_{0.01}$ | $0.66_{0.01}$ | $0.57_{0.09}$ | $0.67_{0.02}$ | $0.77_{0.09}$ | $0.90_{0.01}$ | $0.46_{0.03}$ |
| OR0(NN) | $0.66_{0.02}$ | $0.68_{0.08}$ | $0.66_{0.01}$ | $0.53_{0.05}$ | $0.70_{0.02}$ | $0.84_{0.16}$ | $0.91_{0.00}$ | $0.44_{0.01}$ |
| OR100(NN) | $0.66_{0.01}$ | $0.79_{0.10}$ | $0.65_{0.01}$ | $0.77_{0.11}$ | $0.70_{0.02}$ | $0.85_{0.11}$ | $0.91_{0.01}$ | $0.45_{0.03}$ |
| OR25(NN) | $0.65_{0.01}$ | $0.67_{0.09}$ | $0.66_{0.00}$ | $0.57_{0.07}$ | $0.70_{0.01}$ | $0.79_{0.05}$ | $0.91_{0.01}$ | $0.44_{0.02}$ |
| OR50(NN) | $0.66_{0.01}$ | $0.70_{0.08}$ | $0.66_{0.01}$ | $0.69_{0.19}$ | $0.71_{0.02}$ | $0.85_{0.10}$ | $0.91_{0.01}$ | $0.44_{0.03}$ |
| OR75(NN) | $0.66_{0.02}$ | $0.74_{0.07}$ | $0.66_{0.01}$ | $0.67_{0.09}$ | $0.69_{0.01}$ | $0.83_{0.13}$ | $0.91_{0.02}$ | $0.44_{0.02}$ |

| Method | MEPS19 BA | DI (0.49) | MEPS20 BA | DI (0.49) | Bank BA | DI (0.84) | Adult BA | DI (0.23) |
|---|---|---|---|---|---|---|---|---|
| DU | $0.50_{0.00}$ | $1.00_{0.00}$ | $0.50_{0.00}$ | $1.00_{0.00}$ | $0.50_{0.00}$ | $1.00_{0.00}$ | $0.50_{0.00}$ | $1.00_{0.00}$ |
| NN | $0.67_{0.02}$ | $0.41_{0.05}$ | $0.65_{0.02}$ | $0.41_{0.02}$ | $0.72_{0.07}$ | $0.82_{0.06}$ | $0.61_{0.05}$ | $0.36_{0.22}$ |
| OR0(NN) | $0.69_{0.01}$ | $0.53_{0.03}$ | $0.66_{0.00}$ | $0.51_{0.04}$ | $0.82_{0.02}$ | $0.77_{0.06}$ | $0.63_{0.09}$ | $0.63_{0.32}$ |
| OR100(NN) | $0.69_{0.01}$ | $0.56_{0.04}$ | $0.66_{0.01}$ | $0.53_{0.03}$ | $0.78_{0.07}$ | $0.78_{0.09}$ | $0.74_{0.08}$ | $0.45_{0.22}$ |
| OR25(NN) | $0.68_{0.02}$ | $0.51_{0.03}$ | $0.67_{0.01}$ | $0.52_{0.04}$ | $0.80_{0.02}$ | $0.78_{0.04}$ | $0.70_{0.11}$ | $0.54_{0.31}$ |
| OR50(NN) | $0.69_{0.01}$ | $0.53_{0.03}$ | $0.66_{0.01}$ | $0.55_{0.03}$ | $0.79_{0.05}$ | $0.78_{0.06}$ | $0.73_{0.08}$ | $0.33_{0.05}$ |
| OR75(NN) | $0.68_{0.02}$ | $0.54_{0.03}$ | $0.66_{0.02}$ | $0.54_{0.05}$ | $0.80_{0.03}$ | $0.78_{0.03}$ | $0.69_{0.08}$ | $0.49_{0.25}$ |

