# OpenReview forum: "Oversampling to Repair Bias and Imbalance Simultaneously"
_automl.cc/AutoML/2023/Conference — AutoML 2023 Workshop_

### Official Review · Reviewer_cM3T · 2023-04-07

**Potential Impact On The Field Of Automl Rating:** 4
**Technical Quality And Correctness Rating:** 3
**Clarity Rating:** 4

**Summary Of Contributions:**

The paper introduces a new algorithm, ORBIT, which extends SMOTE as an oversampling technique that is focused on repairing group bias and class imbalance at the same time with tunable parameters. Experiments show that ORBIT successfully repaired performance across many different models and datasets compared to the other techniques.


**Actions Required To Increase Overall Recommendation:**

Adding a limitation section and expanding on the difference in performance will help. Adding some information on the ethical effect of repairing can further strengthen the paper.

**Clarity:**

The paper has been written well.

Figure 2 is an excellent example of the need for balancing bias and class imbalance.

Figure 3 and the step-by-step explanation of the ORBIT algorithm in Section 3 make it easier to read.

There is no explicit limitation section for the specified ORBIT algorithm. For example, all the datasets considered in the study have one or two protected attributes. Does it work for multiple attributes? Also, in some instances, the authors do not provide sufficient explanation. For example, authors claim that ORBIT is very competitive without hyperparameter tuning but why? What are some cases ORBIT can fail and why?. A deep dive will help set up the stage for future work.

**Ethics Details (Optional):**

No ethical issues with the proposed model. However, it would be a bonus to see how the bias mitigator helped repair some ethical issues about the dataset. For example: How much did it improve the prediction on underrepresented classes post-repair?

**Overall Review:**

This a very well-written paper. I have two significant concerns that need to be addressed.

- Add a limitation section to discuss examples/use cases where ORBIT fails.
- Address the difference in reported performance in Figure 4 and Table 4 in [8].


**Potential Impact On The Field Of Automl:**

Group bias and class imbalance are critical to the field of machine learning, especially in order to promote fairness in model predictions. The proposed ORBIT framework can be a great tool for balancing class imbalance and bias at the same time in AutoML systems.

I also encourage the authors to add a sentence or two on how integrating this AutoML system can affect the latency of an ML model.


**Reproducibility (Optional):**

The authors have provided the code with comments and a transparent README file.  Table 4 in [8] shows that the balance accuracy score for SMOTE in the Adult dataset is around ~0.75. But  Figure 4 in this paper shows a Balanced score for SMOTE in the Adult dataset to be ~0.62. Why is there a difference? (Both seem to be for LR and Adult Dataset)

It is unclear why FOS results were not included considering it was mentioned in the related work.


**Review Confidence:**

3: You are fairly confident in your assessment. It is possible that you did not understand some parts of the submission or that you are unfamiliar with some pieces of related work.

**Review Rating:**

7: Weak Accept: Technically sound paper with moderate-to-high impact and strong evaluation, with perhaps some minor flaws.

**Review Summary:**

The ORBIT algorithm can significantly impact AutoML systems to help mitigate the negative effects of group bias and class imbalance simultaneously. The paper is well written and has the potential to be great, considering the authors highlight some of the significant impacts of the algorithm compared to SMOTE, FOS, or other existing techniques.

**Technical Quality And Correctness:**

I appreciate some critical but fundamental considerations the authors have undertaken when evaluating such algorithms. For example:

1. The authors ensured that the training set had been oversampled but did not modify the testing set.
2. The effect of ORBIT is measured by balanced accuracy, which is one of the correct metrics for imbalanced datasets.
3. I agree with the author's recommendations to use stratify split for both groups and classes since. Otherwise, it can tend to produce complex interactions.
4. The authors use cross-validation as intended.

---

> ### Author Response · Authors · 2023-05-01
> **Author response to Reviewer cM3T**
>
> Thank you for your positive review! Regarding your points under "Actions Required To Increase Overall Recommendation:"
>
> > Adding a limitation section
>
> We have added a new limitation section with a discussion of the points you suggested, please see the updated PDF.
>
> > expanding on the difference in performance
>
> You asked why SMOTE on the Adult dataset yielded a different result in the FOS paper than in our paper. We found it difficult to track down the reason. The FOS paper assumes that all features are numeric, but does not discuss how they encode categorical features. We use an ordinal encoder, so it is possible they used a different encoder that yielded better results on this dataset. Furthermore, the FOS paper does not report error bars, so it is even possible that their result simply arose from a lucky split.
>
> > Adding some information on the ethical effect of repairing
>
> You wrote "it would be a bonus to see how the bias mitigator helped repair some ethical issues about the dataset". We ran out of time during the author response period to dig deeper into this. One thought would be to look at individual fairness for samples whose predictions changed. Note that the aggregate effect of such cases is already reflected in the measured disparate impact scores.

---

### Official Review · Reviewer_6LDz · 2023-04-10

**Potential Impact On The Field Of Automl Rating:** 3
**Technical Quality And Correctness:** I believe their experiments and resul…
**Technical Quality And Correctness Rating:** 3
**Clarity Rating:** 4
**Actions Required To Increase Overall Recommendation:** I think the figures should be changed…

**Summary Of Contributions:**

This study introduces a method called ORBIT to mitigate the imbalance and bias at the same time. And they show that the best mitigation always does not lead to the best results, and there should be tuning to achieve the best results.

**Clarity:**

The paper is well-written and presented well. It was easy to follow the content considering the background and explanation provided.

**Overall Review:**

Positive:
- Using an extensive number of data sets and models that help to generalize the idea
- well-presented and written

Negative:
Just two minor issues:

- Why the authors used only binary classification data sets? It would be interesting to see how it generalizes to multi-classification problems
- In Fig 4, 5, 6. for some cases, it is hard to check the results due to the overlapping of texts. Maybe try another way of demonstration or use tables instead?

**Potential Impact On The Field Of Automl:**

Considering its idea of dealing with both imbalance and bias simultaneously to achieve the best possible performance, I think it will be beneficial for the community to integrate such methods into their AutoML algorithms.

**Review Confidence:**

3: You are fairly confident in your assessment. It is possible that you did not understand some parts of the submission or that you are unfamiliar with some pieces of related work.

**Review Rating:**

8: Accept: Technically sound paper with major impact and strong evaluation, with perhaps some minor flaws.

**Review Summary:**

Based on their contribution to making tuning between bias and imbalance and the insights they provided, and also considering the soundness of the paper, I believe having such papers in the community will help the community greatly.

---

> ### Author Response · Authors · 2023-05-01
> **Author response to Reviewer 6LDz**
>
> Thank you for your positive review! Based on your feedback, we have changed all the scatter plots to have non-overlapping data labels.

---

### Official Review · Reviewer_3CZ2 · 2023-04-13

**Potential Impact On The Field Of Automl Rating:** 2
**Technical Quality And Correctness Rating:** 3
**Clarity Rating:** 2

**Summary Of Contributions:**

This paper proposes a simple approach, called ORBIT, that aims to address two types of bias: 1) class imbalance (where some classes have more/fewer examples than other classes); and 2) group bias, which causes examples with certain protected attribute values to be classified into unfavorable outcomes. Class imbalance can reduce the accuracy of a model, while group bias can negatively impact fairness. While oversampling can be used to mitigate both of these types of bias, the level of oversampling is a hyperparameter that should be tuned to balance between achieving good accuracy and fairness---simply reducing class imbalance may increase group bias, and vice versa.

The ORBIT algorithm extends SMOTE (the Synthetic Minority Oversampling Technique). ORBIT has two hyperparameters that can be tuned to focus more on reducing class imbalance or group bias. The paper evaluates ORBIT on 12 datasets, and compares it to three other approaches to mitigate class imbalance and nine other approaches to mitigate group bias. The authors also investigate five different ways to combine metrics for accuracy and fairness, to use single-objective hyperparameter optimization to tune the ORBIT hyperparameters.

**Actions Required To Increase Overall Recommendation:**

I think that the results need to be presented more clearly, and the diagrams (e.g., Figure 3) should be revised to improve clarity. The writing also needs to be reorganized in places to improve clarity, for example in Section 2. Regarding the evaluation, it would be good to have more fairness metrics reported, as was done in prior work such as Fair-SMOTE. The discussion of related work also needs to be substantially improved.

**Clarity:**

The paper has several clarity issues: the plots showing accuracy and disparate impact are very hard to parse, as the labels of many methods overlap; the diagrams (such as Figure 3) are overly complicated; and the organization of the writing needs improvement. Overall, the paper could benefit from another pass over the writing to improve clarity, and the authors should consider alternative ways to visualize the results. Please see the Overall Review box for more details.

**Overall Review:**

**Pros**

* The paper studies an interesting problem, aiming to control the degree to which both class imbalance and group bias are mitigated, by selectively oversampling the data.

* The paper is mostly well-written except for a few important issues discussed in the "Cons" section below.

* The authors provided source code with their submission, with files to reproduce specific figures in the paper. I have not run it, but I believe that the results are reproducible.

* The experimental results (Figures 1, 4, 5, 6) provide error bars, which is nice to see.

* The ORBIT algorithm is very simple; the core of the algorithm simply computes the number of samples to draw for each condition { (Class 0, Group 0), (Class 0, Group 1), (Class 1, Group 0), (Class 1, Group 1)} and then oversamples the data accordingly, using SMOTE.

* Figure 2 (while cluttered and insufficiently annotated) is nice, and does a good job illustrating how oversampling to address only class imbalance may harm group bias, and vice versa. It also shows clearly how ORBIT can repair both class imbalance and group bias, or how it can selectively repair either one without making the other worse.

* The paper performs a decent empirical evaluation, comparing ORBIT to three other approaches for imbalance mitigation and nine other approaches to bias mitigation, across 12 datasets.

* Because there may be a trade-off between accuracy and fairness when mitigating bias, teh authors investigate five metrics using different approaches for combining fairness and accuracy into one objective, that can be used for single-objective (as opposed to multi-objective) hyperparameter optimization.


**Cons**

* The paper deals with only binary classes and binary groups. Is ORBIT only applicable to such binary settings? All of the 12 datasets used in the paper are binary classification tasks.

* The presentation of the results is problematic. In many of the plots, the results overlap so much that the labels are basically unreadable. I think that the authors should consider alternative formats to convey the results, that make it easy to parse each method without needing to guess what the labels are. For example, the results in the Fair-SMOTE paper, presented in tabular form, are relatively easier to parse.

* The caption of Figure 1 is unclear; it does not give enough information to understand the plot (e.g., the metrics used, what each subplot represents, etc.) The titles of all subplots in Figure 1 are identical, which is confusing; these titles would be better changed to "Logistic Regression," "Support Vector Machine," "Gradient Boosting," and "Neural Network" to be more informative about what each subplot shows.

* Section 2 is poorly structured. While the section title is "Problem Statement," it starts with an experimental result rather than describing the problem; it starts with "Figure 1 shows results for ORBIT" before the ORBIT method has been described. In addition, Figure 1 is not clearly described: disparate impact should be defined, and the way in which it is made symmetric should be clarified. The motivation for Figure 1 is to show that "the effect of imbalance and bias repair on estimators is unpredictable," but this is only mentioned at the end of the second paragraph in the section. Either the Figure 1 example should be given after describing the problem (which starts in the third paragraph), or the motivation for Figure 1 should be given before describing the results. For example, it would be better to write an intro sentence like: "To motivate the need to tune the bias repair factor, we first provide an example problem where the effect of bias repair is hard to predict a priori, when training different architectures (logistic regression, SVM, gradient boosting, or a neural network)."

* It seems that only one metric is used to measure fairness: (symmetric) disparate impact (DI). Because the tasks considered for ORBIT are all binary, other metrics are easily applicable, including recall, false alarm, accuracy, precision, and F1 scores---these can be computed from the confusion matrix of binary classification.

* Around Eq. 1, it would be good to clarify whether "disparate impact" is the same as group bias.

* In the paragraph at Line 67, it would be good to state what the optimal values are for the measures (e.g., the optimal class imbalance $\sigma_{\text{ci}}$ and group bias $\sigma_{\text{di}}$.

* For Figure 2, the caption does not explain anything about how to interpret the figure. The numbers on the x- and y-axes are not explained (e.g., 400/600 and 0.333/0.714, etc.) I was able to parse the figure after reading it over for a while, but ideally each component should be described. While I think this figure looks nice, it is quite cluttered; I don't think it is necessary to write the "0_", "1_", "_0", and "_1" labels, because these look confusing without context, and do not add to the figure.

* In theory, can ORBIT always perfectly eliminate imbalance and bias? Are there assumptions under which it can or cannot work?

* In Line 81, it would be good to state the ranges for the hyperparameters $\delta_{\text{imbalance}}$ and $\delta_{\text{bias}}$. That is, are they both in $[0, 1]$?

* Conceptually, it might be good to discuss tasks for which one may only want to eliminate group bias or class imbalance alone, as shown in Figure 2(e,f). If ORBIT can eliminate both types of bias, under what conditions would we only want to eliminate one of them?

* I believe that $n_{\text{ci}}$ and $n_{\text{di}}$ are only defined on Line 82, **after** they were used in the previous paragraph.

* Define "diaeresis labels" when using the term.

* In Step 5 (Lines 141-142), what is the inverse of the get_class function? This step should be explained in more detail.

* In Line 148, instead of "followed by one of the classifiers listed in the caption of Figure 1," it would be clearer to list the classifiers here (logistic regression, SVM, gradient boosting, neural network).

* In Lines 211-213, why does Hyperopt not have the option to select the neural network classifier (only the other three are options)?

* Why is ORBIT not compared to FOS?

* Figure 3 seems overly complicated, especially the "test" branch (why is there a separate node for "X-y split" when this is a dummy operation, as the inputs and targets are separate to start with?). Also, Figure 3 in this paper is quite similar in structure to Figure 4 in Fair-SMOTE, and it is not clear whether this is intentional or not.

* The paper should provide more discussion of the hyperparameters for baseline methods, and how those hyperparameters were tuned.

* The discussion of related work is limited, and could be expanded. I think the paper should give a better overview of different methods to address class imbalance, including undersampling the majority class [1, 2, 3, 4] and oversampling the minority class [1, 5]. These are just a few of the references missing from the paper:

[1] Buda et al., "A systematic study of the class imbalance problem in convolutional neural networks," Neural Networks 2018.

[2] He & Garcia. "Learning from imbalanced data," IEEE 2009.

[3] Japkowicz. "The class imbalance problem: Significance and strategies," AI 2000.

[4] Ouyang et al., "Factors in finetuning deep model for object detection with long-tail distribution," CVPR 2016.

[5] Byrd & Lipton. "What is the effect of importance weighting in deep learning?" ICML 2019.




**Minor**

* In the abstract, it is stated that "This paper demonstrates how to use a Bayesian optimizer to tune ORBIT" --> Why is using BayesOpt here considered a contribution?

* Also in the abstract, it may be slightly misleading to say that "this paper introduces a new bias mitigator along with a methodology for applying automated machine learning for fairness," because this may imply that the notion of using AutoML for fairness is also new, which is not the case.

* Figure 2 in this paper seems similar to Figure 3 in the Fair-SMOTE paper; is this just a different way to visualize a similar setup?

* In Line 33, it is unclear what is meant by "repair the data."

* In Line 34, it should be more clearly explained why the "effect of different mitigation levels is unpredictable." Shouldn't there typically be a predictable effect of the mitigation level with respect to reducing class imbalance at the expense of group bias, or vice versa?

* I think it would be good to use consistent terminology throughout the paper: is "mitigation level" the same as "repair level"?

* In the last paragraph of the introduction, it would be good to clarify which hyperparameters need to be tuned, when discussing AutoML applied to ORBIT.

* I don't think the expanded form of the acronym "ORBIT" is ever written out. I inferred from the title that it is "Oversampling to Repair Bias and Imbalance Together (ORBIT)," but this should be clearly stated in the paper to avoid confusion.

* L75: "highest repair for" --> "highest repair coefficient for"?

**Potential Impact On The Field Of Automl:**

This paper proposes a simple approach for mitigating bias, where the degree to which two different types of bias (class imbalance and group bias) are mitigated is controlled by two hyperparameters. The contributions are loosely related to AutoML; the hyperparameter optimization component is somewhat tacked-on to the paper. It does not make direct methodological contributions to the field of AutoML. The component that is most relevant to this field is the investigation into which method of combining different metrics yields a useful objective for hyperparameter optimization. This paper follows a line of work on AutoML for fairness, and may be cited by future papers in this area.

**Reproducibility (Optional):**

I appreciate that the authors have provided code with their submission. I have not run it, but I believe that the work is reproducible.

**Review Confidence:**

4: You are confident in your assessment, but not absolutely certain. It is unlikely, but not impossible, that you did not understand some parts of the submission or that you are unfamiliar with some pieces of related work.

**Review Rating:**

3: Reject: For instance, a paper with technical flaws, weak impact, and/or weak evaluation.

**Review Summary:**

ORBIT is a simple method that computes new dataset sizes, and uses SMOTE to oversample accordingly. ORBIT provides two hyperparameters that control the degree of mitigation of class imbalance and group bias, respectively. The authors evaluate ORBIT on several binary classification datasets, and compare to several other imbalance and bias mitigators. Unfortunately, the paper suffers from several clarity issues, both in the organization of the writing and in the presentation of results (in particular, the result plots are very hard to read because many points---and their labels---overlap). In addition, the related work is not well covered, and only one fairness metric is reported.

**Technical Quality And Correctness:**

I believe that the method is technically sound, and the results are likely correct. Please also see the detailed comments in the Overall Review box.

---

> ### Author Response · Authors · 2023-05-01
> **Author response to Reviewer 3CZ2**
>
> Thank you for your detailed constructive feedback! We have made extensive changes to the writing of the paper to address as many of your points as we could. Below, we focus on the main points under "Actions Required To Increase Overall Recommendation":
>
> > the results need to be presented more clearly
>
> We have changed all the scatter plots to have non-overlapping data labels.
>
> > the diagrams (e.g., Figure 3) should be revised to improve clarity
>
> We have applied your feedback to improve clarity in the hand-drawn diagrams.
>
> > The writing also needs to be reorganized in places to improve clarity, for example in Section 2.
>
> We have revised Section 2 based on your feedback. Please see the portions highlighted in blue.
>
> > it would be good to have more fairness metrics reported
>
> We can report more fairness metrics in the final version of this paper.
>
> > The discussion of related work also needs to be substantially improved.
>
> We have added a new paragraph at the beginning of the related work citing and discussing general literature on class imbalance correction.

---

### Review · Reproducibility_Reviewer_sCR5 · 2023-04-18

**Completeness Of Code And Dataset Supplement Rating:** 4
**Usability And Ease Of Reproducibility Rating:** 4
**Actions Required To Increase The Reproducibility And Overall Recommendation:** N/A

**Completeness Of Code And Dataset Supplement:**

The code and dataset supplements are complete and sufficient to reproduce the paper’s results.

**Overall Reproducibility Review:**

The readme is easy to understand and easy to follow. I can reproduce the results following the instructions.

**Review Confidence:**

4: You are confident in your assessment, but not absolutely certain. It is unlikely, but not impossible, that you did not understand some parts of the submission or that you are unfamiliar with some pieces of the code or data.

**Review Rating:**

9: Strong Accept, all aspects of this are easily reproducible.

**Review Summary:**

As a reproducibility reviewer, I recommend accepting this paper, since I can reproduce the results in the manuscript.

**Summary Of Necessary Code And Dataset Supplement:**

The author conducted experiments on 12 binary classification datasets to verify their method. For each dataset and configuration, the authors performed a total of six runs, which comprised two repeats of 3-fold cross-validation. The experiment results are presented in three figures, each with 12 subfigures, one for each dataset. In the first figure, the authors show results for several mitigators that either repair only class imbalance or group bias. In the second figure, the authors compare Orbit against nine other bias mitigators from AIF360. In the last figure, the author shows results from using Hyperopt to jointly tune the hyperparameters and select the downstream estimator.

**Usability And Ease Of Reproducibility:**

It is easy to reproduce the results. I can reproduce the results following the README.

---

> ### Author Response · Authors · 2023-05-01
> **Author response to Reproducibility Reviewer sCR5**
>
> Thank you for your positive review!

---

### Official Review · Reviewer_cDaz · 2023-04-18

**Potential Impact On The Field Of Automl Rating:** 1
**Technical Quality And Correctness Rating:** 2
**Clarity Rating:** 3
**Actions Required To Increase Overall Recommendation:** 1. improve motivation
2. improve clar…

**Summary Of Contributions:**

The main contribution of the paper is a multicriteria oversampling method that addresses the problems of bias and class imbalance.


**Clarity:**

Figure 2 presents an excellent visualisation of the scenarios considered.

On the other hand, the presentation of the method is too much focused on implementation, rather than the concepts.

Additionally, the discussion about the realism of synthetic instances is not clear (p4/l135-137) and I'm left with the doubt of whether this is an implementation or conceptual issue.

Some statements are not clear:
-p1/l35
-p1/l39-40
-p3/l88
-p4/l123-125
-p8/l219-220

Finally, is "a pre-estimator bias mitigator that modifies the data used to train downstream estimators" a fancy way of saying pre-processing method?


**Overall Review:**

Although the multicriteria oversampling method that addresses the problems of bias and class imbalance is interesting, it isn't clearly motivated and the relevance for AutoML is arguable.

Additionally, minor issues:
-the statement in p1/l25 is arguable.
-shouldn't the threshold in p5/l163 be a parameter of the evaluation function?


**Potential Impact On The Field Of Automl:**

The paper uses automl for hyperparameter tuning of the proposed method.


**Review Confidence:**

3: You are fairly confident in your assessment. It is possible that you did not understand some parts of the submission or that you are unfamiliar with some pieces of related work.

**Review Rating:**

3: Reject: For instance, a paper with technical flaws, weak impact, and/or weak evaluation.

**Review Summary:**

Besides lack of clarity and many technical issues, the most important reason for my evaluation is that this is not really an AutoML paper.


**Technical Quality And Correctness:**

There are some issues concerning the approach, the experiments and the interpretation of the results.

The motivation should be improved: as pointed out (page 9/lines 238-241), other studies do not find it necessary to address both issues (bias and imbalance) simultaneously. So, stronger motivation for the need should be presented.

Concerning the experimental setup:
-p6/l168-171: the threshold of 80% is completely arbitrary. This would be acceptable in a case study, but not in a paper proposing a new algorithm.
-how are the parameters of Orbit set in the experiments to address RQ1?
-in the related work, several methods are discussed but only one is selected for comparison and no justification is provided. In particular, undersampling, which is a completely different approach, would be interesting.

Finally, concerning the results, it's not clear how the results support the conclusions in:
-p2/l58-59
-p8/l225-226

---

> ### Author Response · Authors · 2023-05-01
> **Author response to Reviewer cDaz**
>
> Regarding your three "Actions Required To Increase Overall Recommendation":
>
> > 1. improve motivation
>
> We have revised Section 2 to make the motivation more explicit. Please see the portions highlighted in blue.
>
> > 2. improve clarity of experimental setup and discussion of results
>
> You were concerned that the 80% threshold is arbitrary. We have made two changes. First, we generalized the definitions of the threshold-based objectives to use \tau instead of a hard-coded value. Second, we added an explanation that the 80% threshold is based on a rule by the US Equal Opportunity Commission.
>
> > 3. complete comparison with state-of-the-art methods
>
> We already compare against 3 imbalance mitigators and 9 bias mitigators. That said, for the final version of the paper, we will add experiments comparing to two additional methods, FOS and Undersampling-multivariate.